

# New Radar Altimetry Datasets of Greenland and Antarctic Surface Elevation, 1991-2012

Maya Raghunath Suryawanshi[1,2], Malcolm McMillan[1], Jennifer Maddalena[2,] Fanny Piras[3], Jérémie Aublanc[3], Jean-Alexis Daguzé[3], Clara Grau[3], Qi Huang[1]

[1]UK Centre for Polar Observation & Modelling, Centre of Excellence in Environmental Data Science, Lancaster Environment Centre, Lancaster University, Lancaster, LA1 4YW, UK
[2]Indian Institute of Science, Bengaluru, India
[3]Collecte Localisation Satellites, 31520, Ramonville, France

*Correspondence to*: Maya Raghunath Suryawanshi (maya2509.surya@gmail.com)

**Abstract.** Over the past three decades, there has been a 4.5-fold increase in the loss of ice from the Greenland and Antarctic Ice Sheets, resulting in an enhanced contribution to global sea level rise. Accurately tracking these changes in ice mass requires comprehensive, long-term measurements, which are only feasible from space. Satellite radar altimetry provides the longest near-continuous record of ice sheet surface elevation and volume change, dating back to the launch of ERS-1 in 1991, and maintained through the successive ERS-2, Envisat, CryoSat-2 and Sentinel-3 missions. To reliably constrain multi-decadal trends in ice sheet imbalance, and to place current observations within a longer-term context, requires continued efforts to optimise the processing of data acquired by the older historical missions, and to evaluate the accuracy of these measurements. Here, we present new ERS-1, ERS-2 and Envisat altimeter datasets, which are derived using consistent and improved retrieval methods, and provide measurements of ice sheet elevation spanning two decades. Through comparison with independent airborne datasets, we provide a comprehensive assessment of the accuracy of these measurements, and the improvements delivered relative to previously available products. These new datasets will be of benefit to a broad range of applications, including the quantification of ice sheet mass imbalance, investigations of the processes driving contemporary ice loss, and the constraint of numerical ice sheet models.

## 1 Introduction

Over the past three decades, the polar ice sheets have substantially increased their contribution to global sea level rise (The IMBIE team, 2018; 2019), with the rate of ice loss expected to accelerate further as Earth's climate warms throughout the 21st century (IPCC, 2019). Our understanding of both past changes and future projections of sea level rise benefit from long-term, multi-decadal observations of ice sheet evolution, in order to quantify historical changes in ice mass and to constrain and validate physical ice sheet models. Such continental-scale datasets are exclusively derived from satellite measurements, with polar-orbiting radar altimeters unique in their provision of near-continuous coverage of both polar ice sheets since the



early 1990's. To date, these altimeters have provided a wealth of information for charting ice sheet evolution, including ice
      sheet topography (Bamber et al., 2009, 2013; Helm et al., 2014; Slater et al., 2018; Otosaka et al., 2019), surface elevation
      changes (Helm et al. 2014; McMillan et al., 2014, 2016; Sorensen et al., 2018; Schroder et al., 2019; Shepherd at al., 2019),
      surface (Slater et al., 2021) and basal (Wingham et al, 2006; McMillan et al., 2013) processes, the location and migration of
      grounding lines (Dawson et al., 2017; Hogg et al., 2018; Konrad et al., 2018) and ice mass imbalance (Zwally et al., 2015,
The IMBIE team, 2018, 2019; Simonsen et al., 2021).

      Radar altimeters were originally developed for ocean applications and, over time, their importance for ice sheet studies has
      been realized (Robin 1966; Wingham et al., 1998, 2006). The earliest high-inclination orbit missions of ERS-1, ERS-2, and
      Envisat all operated in a Low Resolution Mode (LRM), providing a relatively coarse (kilometer-scale) ground footprint, and
      no information relating to the origin of the echo within the ~16 km diameter beam limited footprint. Additionally, onboard
tracking was not optimized for the rugged and highly complex surface topography found around the ice sheets' margins. As
      a result, ice sheet elevation measurements from these historical missions have typically been less accurate than those derived
      from more recent, higher-resolution radar altimeters such as CryoSat-2 and Sentinel-3, and the uncertainties associated with
      these measurements have been less well constrained. This, in turn, has made it more difficult to quantify the longer-term ice
      mass imbalance of Greenland and Antarctica, with certainty, and to place current observations within the context of the
multi-decadal climate record.

      In order to improve the fidelity and useability of measurements arising from these historical missions, episodic reprocessing
      of the altimeter archive is performed. This is designed to allow recent innovations in algorithms and auxiliary datasets to be
      utilized, even for missions which no longer actively acquire data. Until now, the most recent reprocessing of ERS-1, ERS-2
      and Envisat data by the European Space Agency delivered the REprocessing Altimeter Products for ERS-1 and ERS-2
(REAPER) (Brockley et al., 2017), and the Envisat version 3 (Soussi et al., 2018) products. For ERS-1 and -2, REAPER
      integrated a number of improvements into the Level-1 and Level-2 processing chains, most notably the inclusion of the
      retrackers that had been implemented for Envisat processing, new precise orbit solutions, and refinements to the instrument
      calibration (Brockley et al., 2017). For Envisat version 3, improvements were made to a number of the geophysical
      corrections within the Level-2 processing, the definition of the continental ice flag, and the instrument calibration (Casella et
al., 2018). Although these reprocessing activities represented significant advances in product quality at the time of their
      release, they are now more than 5 years old. As such, there is the potential to revisit and refine the algorithms implemented,
      in order to make use of more recent advances in methodology and computational resources. Within this study, which was
      performed within the context of the Fundamental Data Records for Altimetry (FDR4ALT) project funded by the European
      Space Agency, we therefore aim to (1) reprocess the ERS-1, ERS-2 and Envisat archives over both ice sheets, to produce an
improved, time-varying ice sheet elevation dataset spanning the period 1991-2012, (2) perform the most comprehensive
      assessment of measurement accuracy to date, across all missions, using an extensive reference dataset, and (3) develop a
      new, dedicated Ice Sheet Thematic Data Product, which is designed to improve the future useability of this valuable
      historical record.



## 2 Data and Methodology

In this section, we firstly introduce the principal characteristics of each of the satellite radar altimeters utilized in this study, together with the airborne data used for validation purposes. We then describe the methodology used to process the altimetry data, including both the principal Level-2 and Thematic Data Product (TDP) algorithms. Finally, we summarize the approach employed to evaluate the accuracy of these new altimetry datasets.

### 2.1 ERS-1

The ERS-1 satellite was launched in 1991 with an orbital inclination of 98.6°, and was the first altimeter to provide comprehensive coverage of the Greenland and Antarctic ice sheets. Over its lifetime, the mission operated in a number of different phases, with differing lengths of repeat cycle; namely, a repeat cycle of 3 days (03.08.1991 – 04.04.1992 and 23.12.1993 – 10.04.1994), 35 days (04.04.1992 – 23.12.1993 and 21.03.1995 – 10.3.2000) and 168 days (10.04.1994 – 21.03.1995), as detailed in Table 1.  As part of its primary payload, the satellite carried a Ku-band (13.6 GHz) radar

altimeter, named RA. Although the mission finally ended in 2000, the radar altimeter was switched off in June 1996.

The ERS-1 Radar Altimeter operated in two possible tracking modes, "Ocean Mode" and "Ice Mode", corresponding to the two range resolution modes of the instrument. Designed with the purpose of maximizing data retrieval over ice sheet surfaces, the Ice Mode had a number of dedicated characteristics; most notably deploying an increase in the range window width by a factor of four. This increased range window dimension had the impact of reducing the range gate resolution to

approximately 1.82 m, as opposed to 0.45 m for the Ocean Mode. Acquisitions over ice sheets were almost exclusively made in Ice Mode, except for a few cycles of data. For a more extensive description of the specificities of each mode, the reader is referred to Peacock, 1998.

**Table 1.** ERS-1 Orbit Phases


| Name | Start | End | Repeat cycle |
|---|---|---|---|
| **Launch** | 17-Jul-91 | - | - |
| **Commissioning Phase (Phase A)** | 25-Jul-91 | 10-Dec-91 | 3 days |
| **Roll Tilt Mode validation** | 12-Dec-91 | 13-Dec-91 | 35 days |
| **Ice Phase (Phase B)** | 28-Dec-91 | 01-Apr-92 | 3 days |
| **Roll Tilt Mode Campaign (Phase R)** | 04-Apr-92 | 14-Apr-92 | 35 days |
| **Multidisciplinary (Phase C)** | 14-Apr-92 | 21-Dec-93 | 35 days |
| **Second Ice Phase (Phase D)** | 23-Dec-93 | 10-Apr-94 | 3 days |
| **Geodetic Phase (Phase E)** | 10-Apr-94 | 28-Sep-94 | 168 days |
| **Geodetic Phase (Phase F)** | 28-Sep-94 | 21-Mar-95 | 168 days |



| Phase G 2nd Multidisciplinary | 21-Mar-95 | 17-Aug-95 | 35 days |
| Phase G Tandem | 17-Aug-95 | 02-Jun-96 | 35 days |
| Phase G Back-Up | 02-Jun-96 | 10-Mar-00 | 35 days |
| End of mission | 10-Mar-00 | - | - |

## 2.2 ERS-2

The ERS-2 satellite was launched in 1995 and served as a follow-on mission to ERS-1, carrying an identically designed Ku-band radar altimeter (RA) instrument on board. However, unlike ERS-1, the satellite operated a continuous 35-day repeat

cycle throughout the entirety of its lifetime. Although the mission de-orbited in 2011, here we only process data up to June 2003 because the tape recorder failure at that time limited the subsequent geographical coverage to specific regions in close proximity to ground receiver stations (Milligan et al., 2008). Although several National Foreign Stations (NFS) were added between 2003 and 2011, coverage still remained limited, and so we do not attempt to recover data after the tape recorder failure.

## 2.3 Envisat

Envisat was launched in 2002 and carried a dual frequency radar altimeter (RA2), which operated in C-band alongside the traditional Ku-band frequency. Like ERS-2, the mission maintained a 35-day repeat cycle throughout the entirety of its lifetime, which ended in April 2012. One of the most notable advances introduced by RA-2 was the Model Free Tracker (MFT), which was designed to automatically adapt its resolution to the surface type. Envisat thus acquired data in three

different acquisition modes; with High (320 MHz), Medium (80 MHz), and Low (20 MHz) bandwidths (Roca et al., 2009). Figure 1 illustrates the coverage of these modes for a single Envisat cycle, showing that over most of the ice sheet, Envisat provided a much higher range resolution (~0.47 m) than ERS-1 and ERS-2, which both acquired data predominantly in their Ice Mode (with a resolution of ~1.8 m). The exception to this was over the very margins of Antarctica and Greenland, where Envisat provided a similar (~1.9 m) or, at times, worse (~7.5m) range gate resolution. A further distinction between Envisat

and ERS-1/2 was the higher Pulse Repetition Frequency (PRF) of the former. The PRF of Envisat (1795 Hz) compared to ERS (1020 Hz), resulted in a Level-1b waveform that was derived by averaging 100 individual pulses, allowing a greater reduction in radar speckle compared to the 50-pulse onboard averaging performed by ERS-1/2. Finally, Envisat RA-2 waveforms were composed of 128 samples, in comparison to only 64 samples for ERS. This substantially increased the number of range gates that contained useful information, especially within the trailing edge of the echo.





**Figure** 1. Panels a and b. 1°x1° gridded maps showing the distribution of ENVISAT low and medium range resolution mode acquisitions (shaded blue) over Greenland (a) and Antarctica (b), defined as the percentage of data where the measured bandwidth was either 80 MHz or 20 MHz. Panels c and d. 1°x1° gridded maps showing the distribution of the low range resolution mode acquisitions (shaded blue) for ERS-2, defined as the percentage of data where the measured bandwidth was 85 MHz.





## 2.4 Airborne Data

The reference datasets that we use to validate the new ice sheet elevation products are airborne surface elevation measurements acquired by the Airborne Topographic Mapper (ATM) instrument, flown on-board NASA's Operation IceBridge (http://nsidc.org/icebridge/portal/) and pre-IceBridge (https://nsidc.org/data/blatm2) campaigns. Although airborne
campaigns were less frequent over the ERS-1, ERS-2 and Envisat period than during the past decade, these datasets are the most extensive available, and provide a valuable – and largely under-utilized – resource for assessing the accuracy of historical satellite products. For each mission, we therefore identify a cycle that coincides with an extensive airborne campaign, and use this as the basis for our validation activities. In the following paragraphs we describe each of the datasets that were used, in turn.

The Operation IceBridge Airborne Topographic Mapper (ATM) is an airborne scanning LIDAR developed by NASA to map ice surface elevation in the polar regions. Between 2009 and 2020, it was one of the principal instruments carried onboard NASA's Operation IceBridge campaigns. The Level-2 elevation measurements have been resampled to approximately 50 m along-track (varying with aircraft speed) and have a fixed 80 m across-track platelet at aircraft nadir. At a nominal operating altitude (typically 500-750 m above the ice surface) the ATM elevation measurements have been estimated to achieve a
horizontal accuracy of 74 cm, a horizontal precision of 14 cm, a vertical accuracy of 7 cm and a vertical precision of 3 cm (Martin et al., 2012). Although the majority of Operation IceBridge campaigns have been flown since 2010, an extensive campaign was flown in the Spring of 2009, and this dataset is of particular use for validating Envisat elevation measurements.

Prior to the initiation of Operation IceBridge in 2009, NASA's Wallops Flight Facility had flown Greenland airborne
campaigns carrying ATM instruments in nearly every year since 1993. These ATM surface elevation measurements have a similar resolution to the ATM flown onboard Operation IceBridge, although typically offered a lower vertical accuracy of 20 cm (Krabill et al., 1995). We use these measurements to validate our ERS-1 and ERS-2 measurements of surface elevation over Greenland (Figure 2).





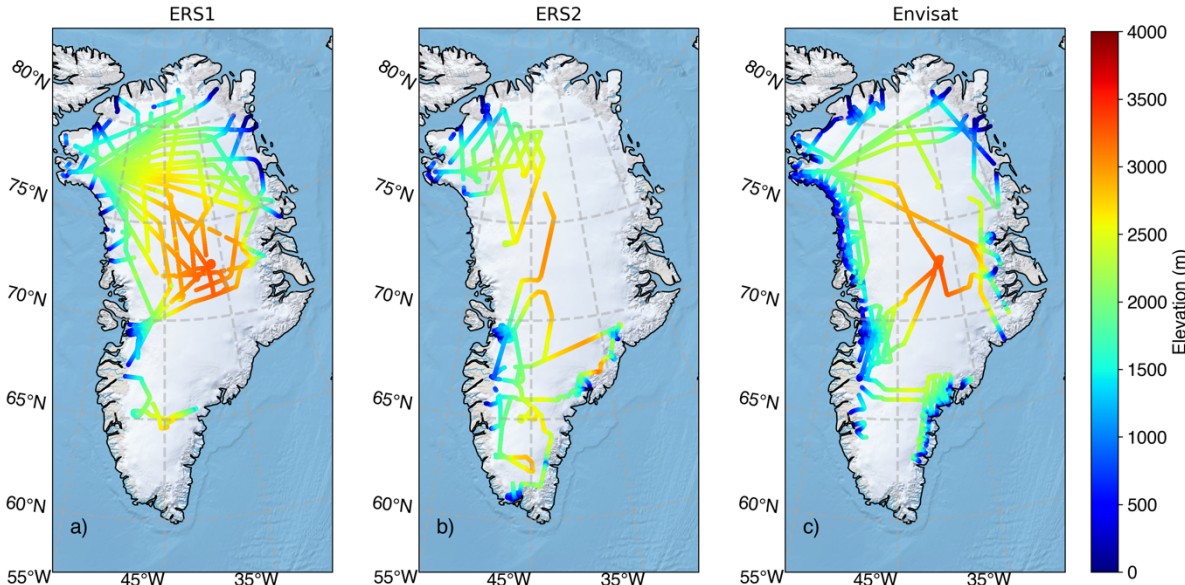

**Figure 2.** The airborne reference datasets acquired over Greenland that were used to validate the new altimetry products, for
a) ERS-1 (airborne campaign during May-June 1994), b) ERS-2 (airborne campaign during May 2003) and c) Envisat
(airborne campaign during March-May 2009).

## 2.5 Altimetry Processing Methodology

### 2.5.1 Level-2 Processing

The primary input data for this study were the ERS REAPER and ENVISAT V3.0 20 Hz waveforms. This section
summarizes the Level-2 processing and corrections that were then applied to estimate ice sheet elevation from these input
products. A complete description of the algorithms is provided in the FDR4ALT Product User Guide (Piras et al., 2023) and
the FDR4ALT Detailed Processing Model Document (The FDR4ALT team, 2023).

First, all ERS and Envisat measurements were selected over ice sheet surfaces, using the ice sheet masks from the
BedMachine dataset (Antarctica v1.38 and Greenland v3.10; Morlighem et al., 2020). For each record, the ice sheet
elevation was calculated as follows:

$$elevation = altitude - altimeter\_range - \sum corr_{geophy} + reloc\_correction, \qquad (1)$$

Where $altitude$ is the satellite altitude from the DEOS solution (Otten, 2019), and from the CNES release "F" of Precise
Orbit Ephemerides (POE) standard (Picot et al., 2018), for ERS and ENVISAT respectively. The $altimeter\_range$ was
derived from the window delay, adjusting for the position of the waveform within the acquisition window via the standard
procedure of waveform retracking, using two different empirical retrackers that were evaluated within the study. The first
retracker was an implementation of the Threshold First Maximum Retracking Algorithm (TFMRA) (Helm et al., 2014), with



a retracking threshold of 25%. The second was the nominal retracking solution from the REAPER and ENVISAT V3.0
processors, which utilizes an OCOG (ICE-1) retracker (Wingham et al., 1986; Bamber, 1994), with a retracking threshold of
30% (Brockley et al., 2017). This retracker is referred to hereafter as the "Threshold Center Of Gravity" (TCOG) retracker,
in recognition of the fact that it is also employs a threshold-based approach. An intercomparison of these two retracking
algorithms is reported in Section 2.6. The *altimeter_range* was also corrected for all instrumental corrections, and
accounts for the different gate resolutions induced by the acquisition modes that were described in Sections 2.1 and 2.3.

To migrate measurements to their estimated origin on the ice surface, we relocated them to the Point Of Closest Approach
(POCA) by adopting the methodology introduced by Roemer et al. (2007), which was implemented using high resolution
Digital Elevation Models (DEM). This represents a significant algorithmic evolution compared to the slope based methods
employed by REAPER and Envisat version 3. The Roemer methodology determines the POCA location by searching for the
minimum satellite-surface range using *a priori* knowledge of the surface topography within the beam footprint, which is
derived from a DEM. *reloc_correction* is the corresponding correction calculated during the relocation processing,
following the formulation specified in Roemer et al. (2007). The auxiliary DEM's employed for the relocation processing
were the Reference Elevation Model of Antarctica (REMA) v1.0 (Howat et al., 2019) and ArcticDEM v3.0 (Porter et al.,
2021) from the Polar Geospatial Center, for measurements acquired over Antarctica and Greenland, respectively.
Measurements where the estimated relocation distance exceeded 20 km from nadir were discarded, and a warning flag was
raised in cases where the relocation distance was between 8 km and 20 km (8 km being the approximate boundary of the -
3dB antenna beamwidth). Figure 2 shows the relocation distances for a single cycle across Greenland and Antarctica.

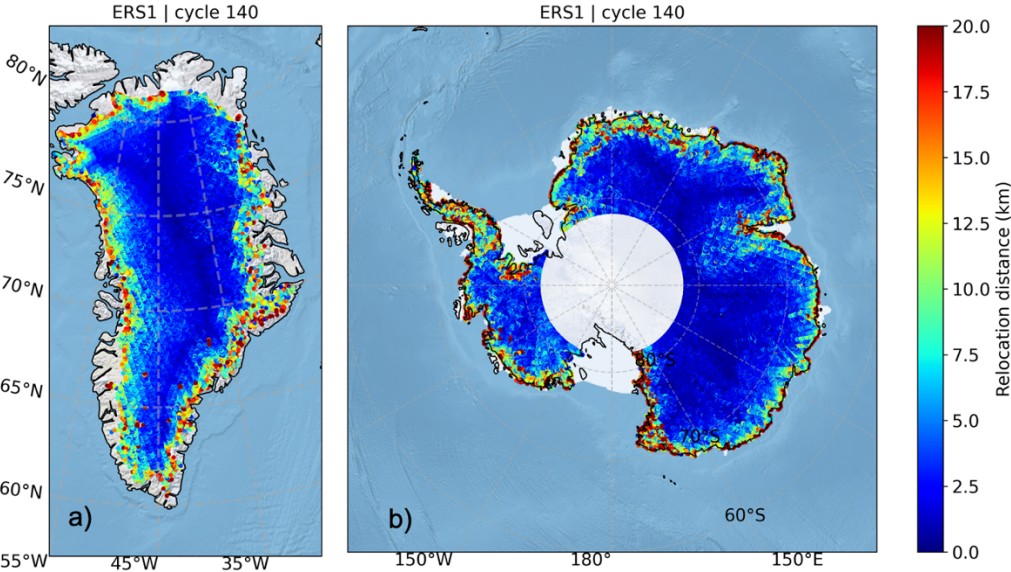

**Figure 3.** Distribution of relocation distance (the distance from satellite nadir to the Point of Closest Approach) across the
(a) Greenland and (b) Antarctic ice sheets for ERS-1 cycle 140.



In addition to the instrument and relocation corrections, a range of standard geophysical corrections were applied,
       Specifically, $\sum corr\_geophy$ denotes the sum of all geophysical corrections applied, and accounts for ionospheric and
       tropospheric delays, and variations in range due to the ocean tide, ocean loading tide, solid Earth tide, pole tide, and the
       inverse barometer effect. Full details relating to the models used are provided in the FDR4ALT Product User Guide (Piras et
       al., 2023).

Finally, two anomalies in the ERS REAPER dataset were identified and corrected. First, the ERS REAPER dataset contains
       sporadic anomalies in the time tag field, which manifest as either a reversal in the normally monotonically increasing nature
       of consecutive measurements, or a jump forward or backward in time. A dedicated algorithm was therefore developed to
       resolve these anomalies (detailed within the FDR4ALT Detailed Processing Model; The FDR4ALT team, 2023), which
       affected on average ~1% of the total dataset (Piras et al., 2023). Second, the presence of negative values in the REAPER

waveform arrays was occasionally found to occur when the backscattered echo power reached high values, typically when
       the reflection originated from a specular surface. These values were corrected (The FDR4ALT team, 2023), which allowed
       an additional 0.3-1.2% of waveforms to be recovered, depending on the mission and the time of the year.

### 2.5.2 Thematic Data Processing Methodology

       Following the Level-2 processing, an additional processing chain was implemented to derive a higher-level Ice Thematic

Data Product (TDP). This chain takes ERS-1, ERS-2 and ENVISAT Level-2 elevation measurements as input, and generates
       a more consistent elevation product at fixed nodes along reference ground tracks. The core methodology is based upon a
       repeat-track processing approach (e.g., Sorensen et al., 2011, Moholdt et al., 2010), whereby data are firstly partitioned into
       along-track segments, and then corrected for the effect of topographic variability within each segment, which arises due to
       the orbital drift of the satellite. These additional steps, which go beyond those of a conventional Level-2 chain, are designed

to deliver a more consistent along-track dataset that maintains the native 20 Hz sampling rate of the altimeter. Additionally,
       we also incorporate uncertainty estimation into the TDP chain, so that each resulting elevation measurement has with it an
       associated uncertainty. In the following text, we summarize the main steps of the TDP chain, and refer the reader to the
       FDR4ALT Detailed Processing Model (FDR4ALT team, 2023) for full details of the algorithmic implementation.

       For each cycle, all Level 2 measurements acquired over Greenland and Antarctica are ingested, and waveform quality flags

and echo relocation flags are applied to remove poor quality records. An additional filter is applied to remove elevations that
       differ by more than 100 m from a reference DEM, specifically ArcticDEM for Greenland (Porter et al., 2018) and REMA for
       Antarctica (Howat et al., 2019). Typically, this filter removes at most 2 % of the ingested data.

       Next, a reference ground track is defined for each satellite pass, based upon the cycle which has a start point that is closest to
       the median of all start points of that pass. This reference track is then sampled at ~ 380 m intervals to create the reference

nodes that form the common basis of the TDP product. For each reference track, a rectangular search window around each
       node is calculated, which has an along-track dimension equivalent to the reference node spacing (~380 m) and an across-



track dimension (20 km) that is chosen to cover the maximum Level 2 relocation distance plus a buffer to account for the across-track orbit drift. This ensures that the search window will encapsulate all POCA measurements, irrespective of how far they have been migrated in the Level-2 echo relocation step.

For each search window along the satellite track, data from all cycles that fall within that search window are identified and associated with the respective reference node. In areas of high topographic relief, more than one POCA measurement per cycle can be segmented in a single search window. In this instance, the POCA measurement that is closest in elevation to its reference node (as determined using the DEM) is selected. In cases where multiple POCA measurements lie in a single search window and these POCA locations (and hence elevations) are identical, we cannot select a single measurement based

upon the aforementioned criteria, and hence the median of the elevation measurements is calculated.

Next, all POCA points for all cycles are migrated onto their associated reference nodes. In essence, this step corrects for the topographic difference in elevation between each POCA measurement location and the reference node location. The topographic difference in elevation is computed using a relatively coarse resolution version of the ArcticDEM (500 m) and REMA (200 m) products, so as to broadly align with the resolution of the pulse limited altimeter footprint. More specifically,

for each POCA elevation, $z(i, j)$, at a location $i, j$, then the migrated elevation $z(i', j')$ at the associated reference node location $i', j'$ is given by:

$$z(i', j') = z(i, j) + \Delta z_{topo}(i, j), \tag{2}$$

Where,

$$\Delta z_{topo}(i, j) = z_{DEM}(i', j') - z_{DEM}(i, j), \tag{3}$$

and $z_{DEM}(i, j)$ and $z_{DEM}(i', j')$ are the DEM elevations at the POCA location and the reference node, respectively. Around the very margin of the ice sheet, in areas of extremely rugged topography, occasionally this topographic correction becomes unreliable. Therefore, if the magnitude of the topographic correction exceeds 200 meters, then the correction is deemed

unreliable and the corresponding elevation at the reference node is set to the fill value. Less than 1% of the topographic corrections exceed 200 m ice sheet wide, for both ice sheets, and for all cycles.

The uncertainty associated with each elevation measurement is estimated using an empirical parameterization, based upon elevation differences between near co-located (within 500 meters) and near coincident (within 30 days) satellite and airborne

measurements. Specifically, uncertainty is parameterized as a function of surface slope. This is motivated by the knowledge that measurement accuracy degrades as a function of ice sheet surface slope, due to the challenges of retracking waveforms and identifying the correct echoing point over increasingly complex ice sheet terrain. More specifically, uncertainty is determined independently for each satellite mission by implementing the following processing steps. First, pairs of near co-located, near co-incident airborne and satellite data are corrected for residual topographical differences due to any

differences in location. Then satellite-minus-airborne elevation differences are computed for each pair. Next, an estimate of



the magnitude of the surface slope at each comparison point is retrieved from an auxiliary DEM. An uncertainty look-up table is then defined by collating individual satellite-minus-airborne elevation differences within 0.1° slope bands and computing the median of the absolute elevation differences within each slope band. These median values typically lie in the range 0-10 m, depending upon the magnitude of the slope and the satellite mission. At high slopes (> 1.3°) the number of

comparison points becomes small (< 100 measurements per slope bin, on average; compared to 102-103 measurements per bin at lower slopes). As a result, the statistics for higher slope bins are relatively unstable, with a standard deviation typically 2-4 times higher. For slopes greater than 1.3 degrees, we therefore assign uncertainties based upon a linear regression of uncertainty against slope with 0 intercept, which is fitted to the uncertainties of the lower slopes (< 1.3 degrees). Finally, for each altimetry measurement, the surface slope is estimated at its echoing location and then, the mission-specific look up

table is used to assign an associated uncertainty to that altimetry measurement.

### 2.5.3 Neural Network Classification

It is well established (McMillan et al., 2021; Huang et al., 2024) that rugged ice sheet topography can introduce complexity into the shape of the returned waveform, which in turn can complicate the retrieval of estimates of ice sheet elevation. As such, analyzing the morphologies of waveforms acquired over ice sheets can inform our understanding of the reliability of

retrievals from different waveform classes. To this end, an existing supervised neural network classifier (Poisson at al., 2018) was first used to discriminate different Envisat Ku-band waveform shapes. Specifically, this algorithm has been designed to predict the most likely class for each echo, based upon a subset of possible classes, which are taken from a global reference dataset spanning all surface types and missions (Table 2 and Table 3). Following Poisson et al., 2018, we selected 12 classes (1 to 9, 12 and 16) for the purposes of classifying and differentiating different types of Envisat waveform.


**Table 2**. Schematic drawings of the main waveform classes within the global reference dataset; descriptions of each class are provided in Table 3.

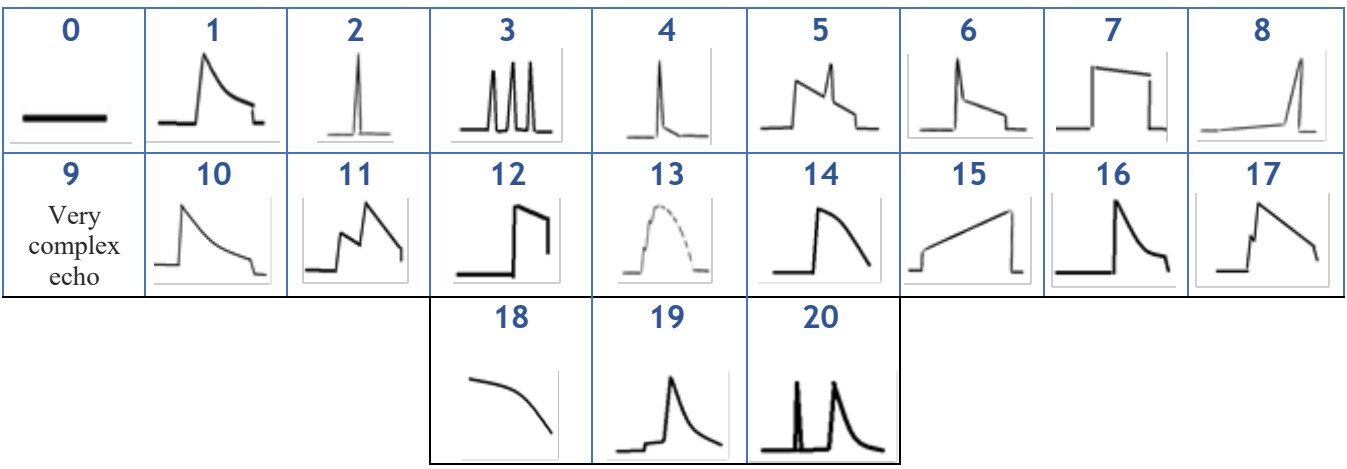



**Table 3**. Brief descriptions of the main waveform classes within the global reference dataset.

| | |
|---|---|
| **1** | Brownian |
| **2** | Highly specular |
| **3** | Multiple peaks |
| **4** | Moderately specular |
| **5** | Brownian with a peak on the trailing edge |
| **6** | Brownian with a peak on the leading edge or a steep trailing edge |
| **7** | Brownian with a flat trailing edge |
| **8** | Strong peak at the end of the analysis window |
| **9** | Very complex echo |
| **10** | Brownian with high thermal noise |
| **11** | Double leading edge |
| **12** | Shifted Brownian |
| **13** | Brownian with a disturbed leading edge |
| **14** | Volume-Brownian |
| **15** | Linear rise |
| **16** | Right-shifted Brownian waveform |
| **17** | Breakage on the leading edge of a Brownian waveform |
| **18** | Linear decrease |
| **19** | Small step before leading edge |
| **20** | Peaky echo before Brownian echo |


Building upon the work of Poisson et al., 2018, we also extended the same methodology for use with ERS-1 and ERS-2, developing a new supervised neural network that was trained and validated using radar waveforms from these missions. First, the learning step consisted in labelling thousands of waveforms (see Table 4**)** that were acquired over different surface types and in different operating modes, in order to capture the full variety of surface slope, roughness and backscattering

characteristics. This step utilized data from January 1996 for both ERS-1 and ERS-2 (respectively cycle 8 and cycle 153), and allowed us to determine the subset of classes applicable to the RA altimeter, according to Table 2. Due to the differences in instrument design between RA and RA-2 (as detailed in Section 2.1), which impact upon the waveform range resolution and shape, the relevant classes are not identical, and we find that for RA the most relevant classes are 1-7, 9-11, 13 and 15-18.

The second step was to determine the set of geometrical parameters describing the RA waveform, to be used as input to the neural network. Indeed, we did not consider the whole waveform as input but the following set of 11 waveform parameters:



(1) leading edge slope, (2) trailing edge slope, (3) thermal noise slope, (4) amplitude of the main peak on the trailing edge and (5) the thermal noise, (6) a breakage flag on the leading edge, (7) the centre of gravity of the waveform, (8) the mean square error between a mean ocean waveform and each measurement, and (9) the global peakiness, (10) kurtosis, and (11) skewness. To assess the performance of the neural network in predicting classes, the collected dataset was split into two strictly independent subsets for training (10,616 waveforms) and testing (3,982 waveforms) representing respectively 73% and 27% of the manually collected dataset (Table 4). We considered ocean class 1 to be the most represented class (46% of the dataset), as oceans cover most of the Earth's surface, but not more than 50% to avoid over-fitting.

**Table 4:** The number of labelled waveforms for each class for both training and testing datasets.

| Class | 1 | 2 | 3 | 4 | 5 | 6 | 7 | 9 |
|---|---|---|---|---|---|---|---|---|
| Training | 4723 | 363 | 240 | 1096 | 710 | 1648 | 252 | 279 |
| Testing | 2023 | 121 | 80 | 365 | 236 | 549 | 84 | 92 |
| Class | 10 | 11 | 13 | 15 | 16 | 17 | 18 | |
| Training | 153 | 160 | 212 | 132 | 189 | 312 | 147 | |
| Testing | 51 | 53 | 70 | 43 | 63 | 103 | 49 | |

The network itself was designed as a feed-forward single-layer neural network and was built using the R package *nnet* (Venables and Ripley, 2002), with 11 neurons in the input layer, 25 in the hidden layer and 15 as outputs. We used a *softmax* activation function in the output layer to generate the probability that a given waveform belongs to the respective classes, and a decay value of 1e-12 to prevent from overfitting. The performance of the neural network classifier was assessed using the test database, and results are presented in Table 5 for the most dominant classes over ocean and ice regions. Full details of the algorithm are provided in the FDR4ALT Detailed Processing Model (FDR4ALT team, 2023) and Product Validation Report (Piras, 2023) documents.

**Table 5**. Performance of the neural network classifier on the test database for ocean Brownian waveforms (class 1), peaky waveforms (classes 2 and 4) and sharp Brown-like waveforms (class 6).

| Class | 1 | 2 | 4 | 6 | others |
|---|---|---|---|---|---|
| Success (%) | 95.4 | 87.6 | 88.0 | 88.4 | 60 |
| Failure (%) | 4.6 | 12.4 | 12.0 | 11.6 | 40 |
| Total (%) | 100 | 100 | 100 | 100 | 100 |



## 2.6 Level-2 Validation Methodology

To evaluate the accuracy of the newly processed altimetry datasets, we computed elevation differences relative to co-located, cotemporaneous airborne data. Specifically, for each satellite mission we evaluated elevation measurements derived from three different altimeter processing configurations; the existing REAPER (Brockley et al., 2017) and Envisat version 3 (Soussi et al., 2018) products available from the European Space Agency, together with output from the new FDR4ALT processing chain, which included two different retracking solutions. Further details are mentioned in Table 6. This

intercomparison was designed to provide a systematic benchmarking of each of the new datasets against existing products, to assess whether they offered an improvement in measurement accuracy.

To evaluate each processing configuration, we first identified all airborne measurements acquired within a 500-meter search radius (i.e. less than half the altimeter's pulse limited footprint) and within 30 days of each satellite measurement. We then used these to compute elevation differences (satellite-minus-airborne) between each pair. This method is similar to that

reported in McMillan et al. (2019), except that here we also introduced a temporal constraint on the search, to avoid the need to correct for temporal changes in elevation. To limit the inclusion of anomalous data, we removed airborne elevation measurements that are greater than 5000 meters, applied retracking (ICE-1, TFMRA) quality flags and Roemer relocation slope correction flag (limiting the relocation distances up to 20 km) to the altimeter products, and removed outlying altimetry measurements that deviated by more than 100 m from a reference DEM. We then corrected for the effect of topographic

variations within the 500 m search radius using an auxiliary DEM. Finally, we calculated estimates of the overall bias (median) and dispersion (MAD; Median Absolute Deviation from the median) of elevation differences for each mission relative to the reference data, and the proportion of outliers (defined as elevation differences with a magnitude exceeding 10 meters). We also investigated the relationship between the magnitude of the elevation differences and geophysical parameters such as surface slope and altimeter waveform shape.


**Table 6.** Satellite data and processing configurations used in the comparison with airborne data.

| Processing Configuration | Satellite | Relocation method | Retracker | Validation Cycle | Validation Date |
|---|---|---|---|---|---|
| **E1 REAPER** | ERS-1 | Linear slope (Brockley et al., 2017) | ICE-1 (Brockley et al., 2017) | 140 | 5/1994 – 9/1994 |
| **E1 ROEMER + TCOG** | ERS-1 | Roemer (Roemer et al., 2007) | TCOG (Brockley et al., 2017) | 140 | 5/1994 – 9/1994 |



| | | | | | |
|---|---|---|---|---|---|
| **E1 ROEMER + TFMRA** | ERS-1 | Roemer (Roemer et al., 2007) | TFMRA (based on the algorithm definition of Helm et al., 2014) | 140 | 5/1994 – 9/1994 |
| **E2 REAPER** | ERS-2 | Linear slope (Brockley et al., 2017) | ICE-1 (Brockley et al., 2017) | 84 | 4/2003 – 6/2003 |
| **E2 ROEMER + TCOG** | ERS-2 | Roemer (Roemer et al., 2007) | TCOG (Brockley et al., 2017) | 84 | 4/2003 – 6/2003 |
| **E2 ROEMER + TFMRA** | ERS-2 | Roemer (Roemer et al., 2007) | TFMRA (based on the algorithm definition of Helm et al., 2014) | 84 | 4/2003 – 6/2003 |
| **EV GDR V3** | Envisat | Linear slope (Soussi et al., 2018) | ICE-1 (Soussi et al., 2018) | 78 | 4/2009 – 5/2009 |
| **EV ROEMER + TCOG** | Envisat | Roemer (Roemer et al., 2007) | TCOG (Brockley et al., 2017) | 78 | 4/2009 – 5/2009 |
| **EV ROEMER + TFMRA** | Envisat | Roemer (Roemer et al., 2007) | TFMRA (based on the algorithm definition of Helm et al., 2014) | 78 | 4/2009 – 5/2009 |

## 3. Results and Discussion

In this section, we firstly assess the coverage offered by our new FDR4ALT datasets. Next, we evaluate the accuracy of the Level-2 elevation measurements with respect to airborne reference data, and compare this to the accuracy of the pre-existing
REAPER and Envisat version 3 products. We then assess waveform morphology over the ice sheets, using our Neural Network classification to investigate the impact of waveform morphological type upon the accuracy of the elevation retrievals. Finally, we analyze the characteristics of the Thematic Data Product, which applies additional processing steps that are designed to improve the homogeneity of the data for the end user.





## 3.1 Spatial Coverage

First, we evaluated the spatial coverage provided by each mission and the different processing scenarios. For this analysis, we selected one 35-day repeat cycle for each mission; cycle 155 (24.3.1996 to 28.4.1996) of ERS-1, cycle 77 (26.08.2002 to 30.09.2002) of ERS-2 and cycle 78 (06.04.2009 to 11.05.2009) of Envisat, to ensure a comparable orbital sampling pattern across all missions. An overview of the coverage is shown in Figure 4. Although coverage is broadly comparable across all processing scenarios, close to the ice margin the new FDR4ALT solutions exhibit less continuous along-track sampling, due to the Roemer approach to echo relocation (Roemer et al., 2007). This is expected to be more realistic in complex topographic regions such as this, where the point of closest approach is sensitive to smaller wavelength topographic features within the altimeter beam footprint, in addition to the large-scale slope.

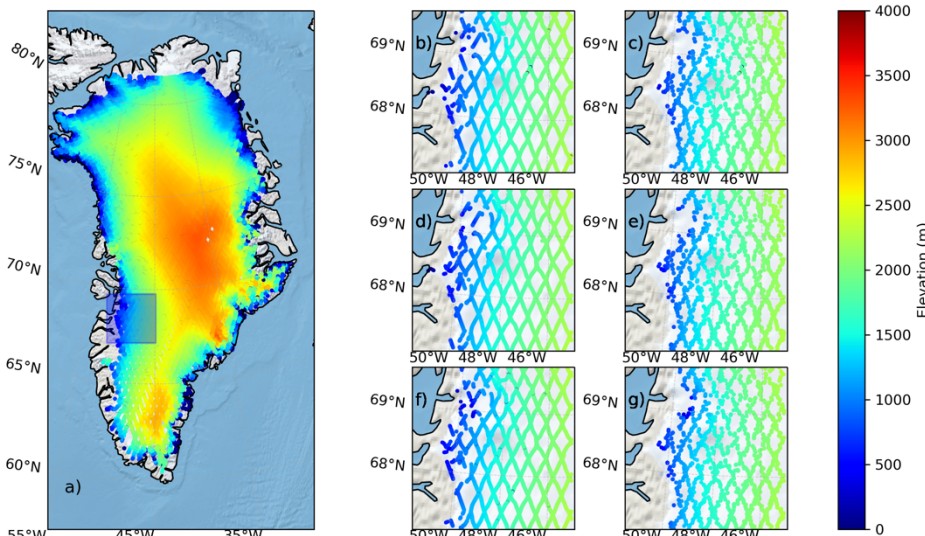

**Figure 4.** Comparison of the coverage of elevation measurements provided by different processing configurations and missions over the Greenland Ice Sheet (panel a) and the Russell Glacier region of Western Greenland (panels b-g). Panel a. The coverage provided by FDR4ALT with TCOG retracking for Envisat cycle 78. Panels (c) to (g). The coverage provided over the Russell Glacier region in Western Greenland (blue box marked on (a)) by ERS-1 REAPER (b) and FDR4ALT TCOG (c) cycle 155; ERS-2 REAPER (d) and FDR4ALT TCOG (e) cycle 77; and Envisat version 3 (f) and FDR4ALT TCOG (g) cycle 78.

Secondly, we assessed the sampling provided by each mission within different bands of ice sheet surface slope (Figure 5), by computing the proportion of 2 x 2 km grid cells that contained at least one valid elevation measurement. Within the low-slope interior of the ice sheet, the 35-day orbit yields ~ 7-8% coverage of grid cells, and this progressively decreases with





higher slope. By the time slope exceeds 1.5° and are lesser than 2° (constituting ~6% of ice sheet grid cells), for example, coverage is typically below 2% for all missions. It is important to note, however, that the percentages are dependent upon the size of grid used, and using a coarser resolution grid would result in higher percentage values. Here we choose 2 x 2 km, to be approximately equivalent to the size of the pulse limited altimeter footprint. Comparing the relative coverage provided by the three missions, we find that overall, they are similar, albeit at higher slopes there is a small progressive improvement

from ERS-1, to ERS-2, and to Envisat (Figure 5).

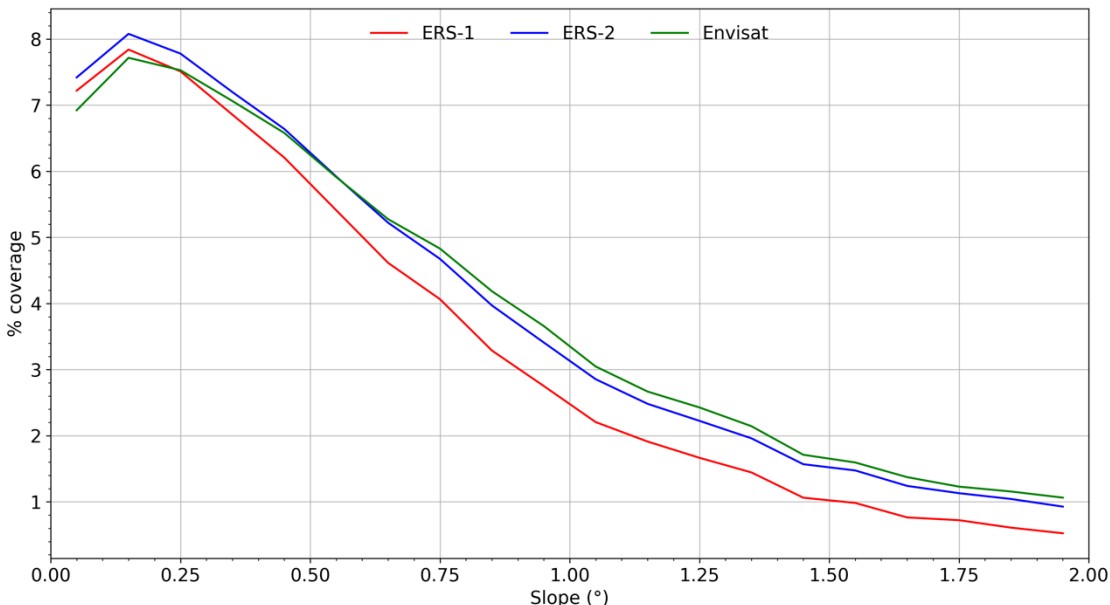

**Figure 5.** The coverage provided by each mission over the Greenland Ice Sheet as a function of ice sheet surface slope. The coloured lines represent the percentage of 2 km grid cells sampled by at least one valid measurement from ERS-1 cycle 155, ERS-2 cycle 77 and Envisat cycle 78.


## 3.2 Level-2 Accuracy Assessment

### 3.2.1 Envisat

We assessed the accuracy of both the existing Envisat version 3 product and the new FDR4ALT Level-2 datasets (the latter of which includes two retracking solutions; Threshold Centre of Gravity retracking and TFMRA retracking), through

comparison with our airborne reference dataset. The spatial pattern of elevation differences is shown in Figure 6 and the overall distributions of the satellite and airborne elevations is shown in inset of Figure 6. Whilst all solutions perform well within the lower slope interior of the ice sheet, it is clear that both of the FDR4ALT solutions exhibit a reduced number of large outliers in regions close to the ice margin. This contrasts with the existing version 3 product, which commonly exhibits



elevations that deviate by more than 10 meters from the coincident airborne data (Figure 6a). This is reflected in the large (~
20 %) reduction in the percentage of FDR4ALT comparison points classified as outliers (~14 %), as compared to the
equivalent statistic for the version 3 product (~35 %; Table 7). Even within the interior of the ice sheet, improvements are
evident in the FDR4ALT solutions relative to GDR version 3, with a number of the inland tracks exhibiting fewer outliers.
Overall, there are significant reductions in both the bias and the dispersion statistics for the FDR4ALT products, with the
FDR4ALT elevation bias reduced by 78 % relative to version 3, and the dispersion reduced by 67 % relative to version 3
(Table 7).

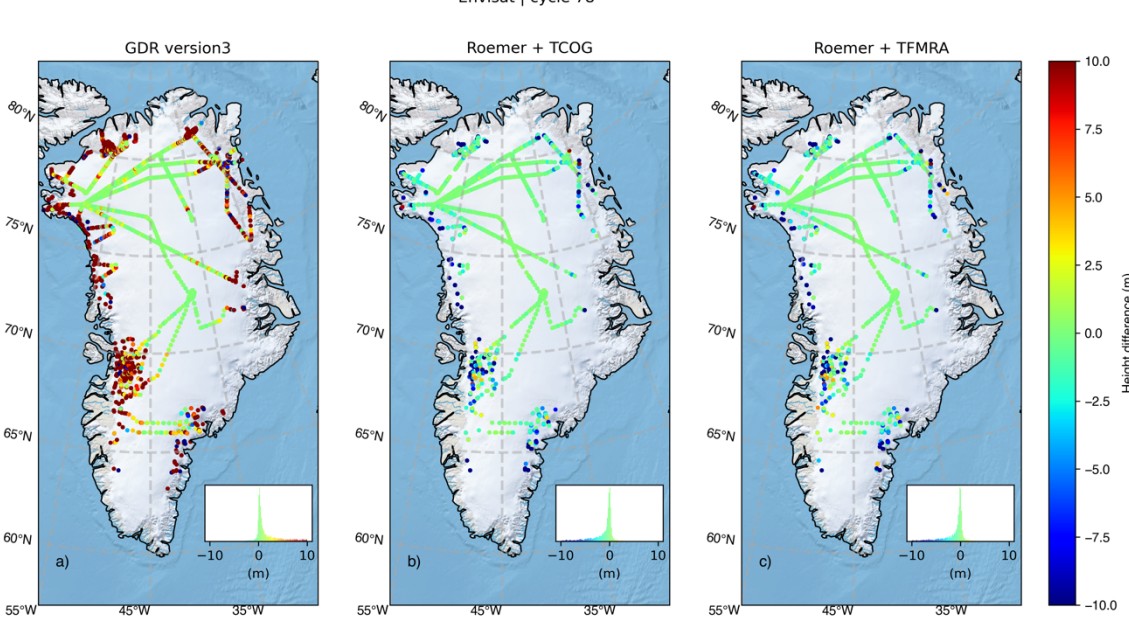

**Figure 6.** Comparison of elevation differences (Envisat minus airborne) over the Greenland Ice Sheet, for the different
processing configurations; the baseline version 3 product (panel a), the FDR4ALT Roemer + TCOG configuration (panel b),
and the FDR4ALT Roemer + TFMRA configuration (panel c). In each panel, the insets show the distribution of elevation
differences.

Figure 7 presents the same comparison data as density scatter plots, and illustrates the level of agreement between the
retrieved satellite elevations and the coincident airborne elevations (panels a-c). This assessment shows that, at low
elevations, the existing version 3 product suffers from an increasingly positive bias (evident from the divergence away from
the 1-1 line). In contrast, due to a combination of the more sophisticated echo relocation and more stringent quality control,
neither of the two FDR4ALT configurations shows the same artefacts. Because the overall variance in ice sheet elevation
due to its topography (~ 0-3500 m) is much larger than the variance between the altimeter and airborne measurements, we
also use an auxiliary Digital Elevation Model to remove the large-scale topographic variation that is common to both
datasets. This isolates more clearly the residual differences between the airborne and altimeter measurements (Figure 7d-f).



This shows that there are a greater number of positive elevation artefacts present in the version 3 dataset (Figure 7d), that are
absent from the FDR4ALT solutions (i.e. the power rising vertically upwards from the origin in panel d). Finally, comparing
the two FDR4ALT configurations (TCOG and TFMRA) indicates that similar results are achieved for both retrackers.
Overall, there is less than 0.02 m difference in both the bias and the dispersion, although TCOG does slightly reduce the
number of large outliers, with a 1.3 % reduction in the number of comparison points deviating by more than 10 m from the
airborne measurement (Table 7).

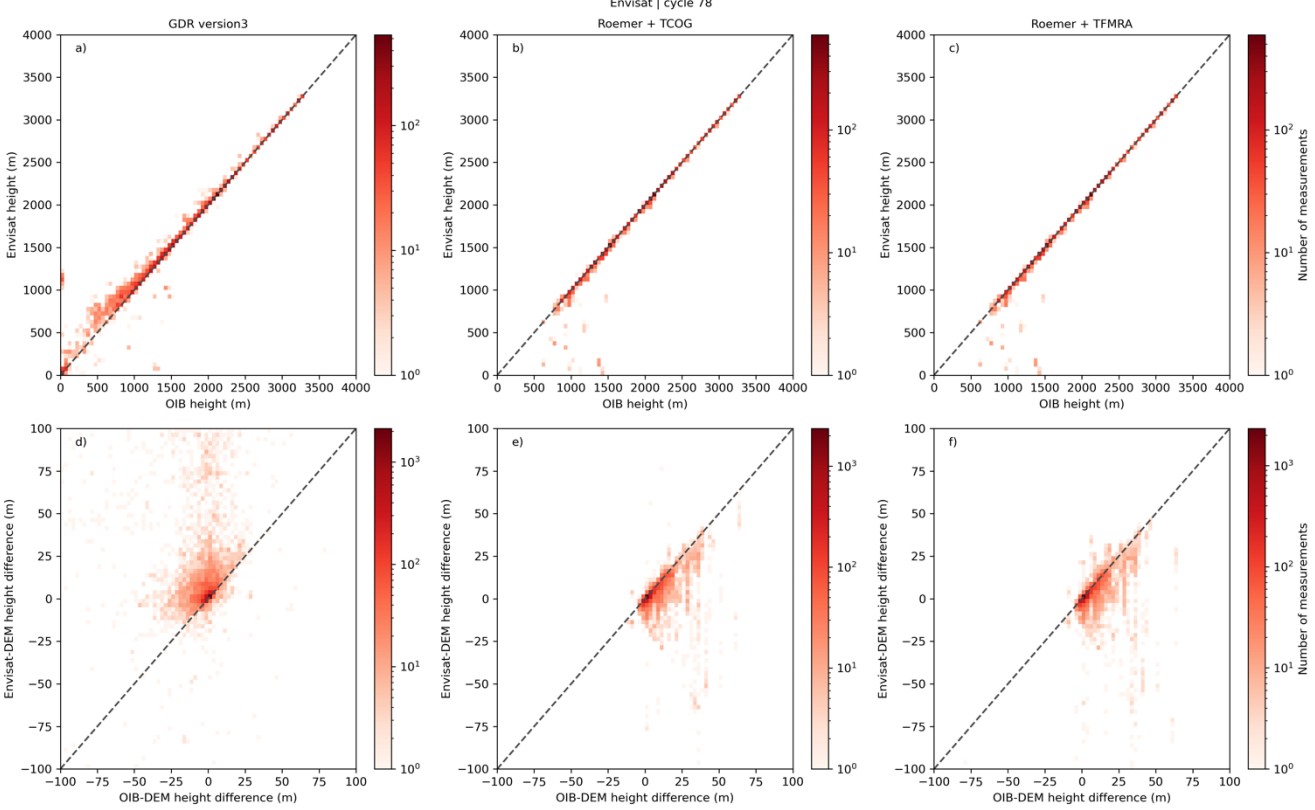


**Figure 7.** Density scatter plots showing the distributions of elevation measurements recorded by the Envisat and Operation
IceBridge (OIB) airborne platforms. The top row (panels a-c) shows the original elevations; the bottom row (panels d-f)
shows elevation residuals relative to an auxiliary DEM, which is used to remove the largescale topographic variance.

### 3.2.2 ERS-2

Next, we performed the same comparative analysis for ERS-2. Figure 8 shows maps of ERS-2 elevation differences with
respect to the reference data, and Figure 9 presents density scatter plots of their respective distributions. As was the case for
Envisat, whilst all solutions again perform relatively well within the lower slope interior of the ice sheet, both FDR4ALT
datasets exhibit a reduced number of outliers close to the margin. This is evident as a 7 % reduction in the percentage of



FDR4ALT comparison points that have an absolute elevation difference greater than 10 m (9.5 % for FDR4ALT, compared
to 16.8 % for REAPER; Table 7). In terms of the central part of the distribution, we again find that the FDR4ALT solutions
outperform REAPER, most notably providing a 73 % reduction in the magnitude of the bias (for example, from +0.93 m to -
0.25 m for the FDR4ALT TCOG solution). Additionally, the FDR4ALT solution also yields a modest (3.4 %) improvement
in the MAD dispersion (Table 7).

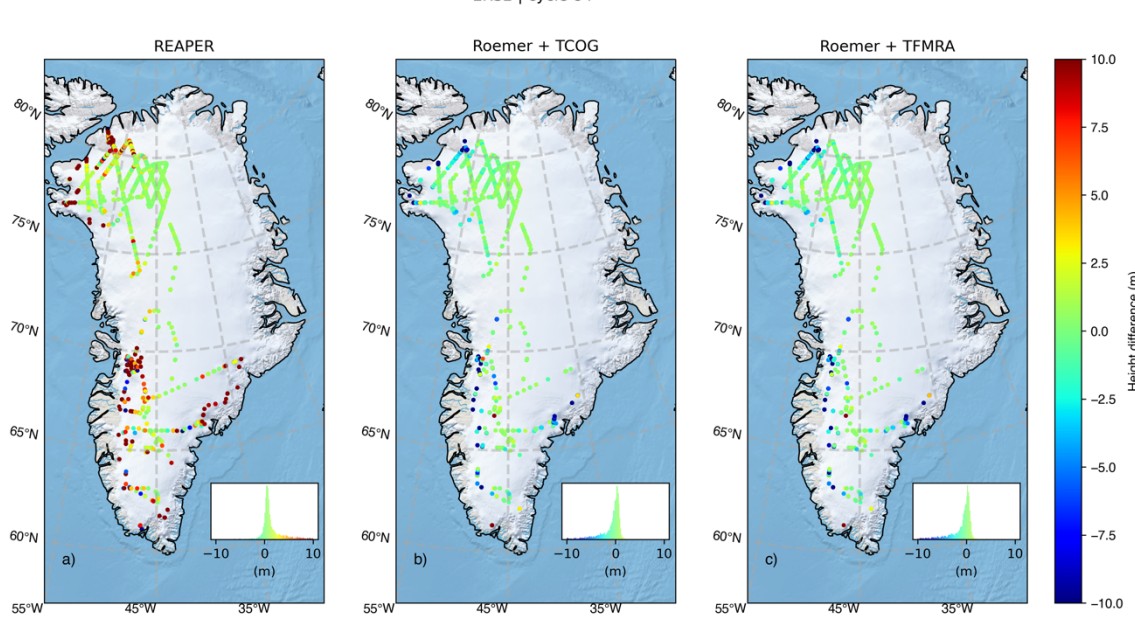

**Figure 8.** Comparison of elevation differences (ERS-2 minus airborne) over the Greenland Ice Sheet, for the different
processing configurations; the REAPER product (panel a), the FDR4ALT Roemer + TCOG configuration (panel b), and the
FDR4ALT Roemer + TFMRA configuration (panel c). In each panel, the insets show the distribution of elevation
differences.

Figure 9 shows the same comparison data, displayed as density scatter plots that compare the retrieved satellite elevations
and the coincident airborne elevations (panels a-c). This shows a similar pattern to Envisat, with REAPER exhibiting an
increasingly positive bias at lower elevations, albeit the divergence is not as pronounced as was the case for Envisat (Figure
7). Mirroring the results for Envisat, the FDR4ALT solutions successfully remove this artefact. Comparing the two
FDR4ALT configurations themselves, again shows relatively small differences between the two retracking solutions (Table
7); albeit with the TCOG solution showing modest improvements in terms of reducing both the magnitude of the elevation
bias (10 cm improvement) and the MAD (5 cm improvement).





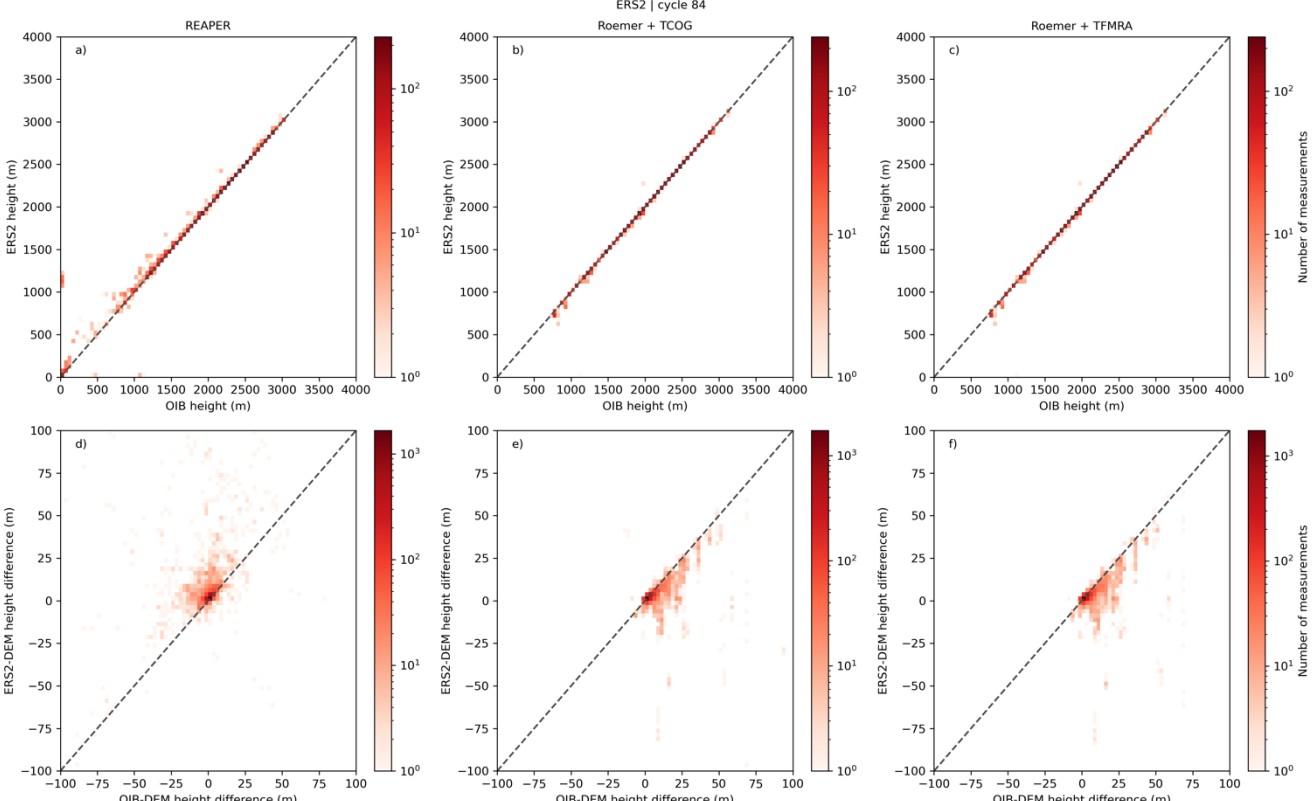

**Figure 9.** Density scatter plots showing the distributions of elevation measurements recorded by the ERS-2 and Operation IceBridge (OIB) airborne platforms. The top row (panels a-c) shows the original elevations; the bottom row (panels d-f) shows elevation residuals relative to an auxiliary DEM, which is used to remove the largescale topographic variance.

### 3.2.3 ERS-1

Finally, we performed the same analysis for the ERS-1 processing configurations. Figure 10 shows maps of ERS-1 elevation differences with respect to the reference data, and Figure 11 presents density scatter plots of their respective distributions. Comparing the statistics (Table 7), we find that although the FDR4ALT solutions exhibit a larger bias than the REAPER solutions, they deliver both a modest reduction in the MAD dispersion (a 6 % reduction for the TCOG solution), and the proportion of outliers (from 14 % to 11 %). The density scatter plots shown in Figure 11 demonstrate similar behavior to Envisat and ERS-2, with REAPER exhibiting an increasingly positive bias at lower elevations. As was the case for both Envisat and ERS-2, the FDR4ALT solutions successfully removed this artefact. Finally, comparing the two FDR4ALT configurations shows that the TCOG solution offers a slightly lower dispersion (0.76 cm vs 0.81 cm, for TCOG and TFMRA, respectively), whereas the TFMRA solution delivers a lower magnitude bias (-0.81 cm vs -0.36 cm, for TCOG and TFMRA, respectively).



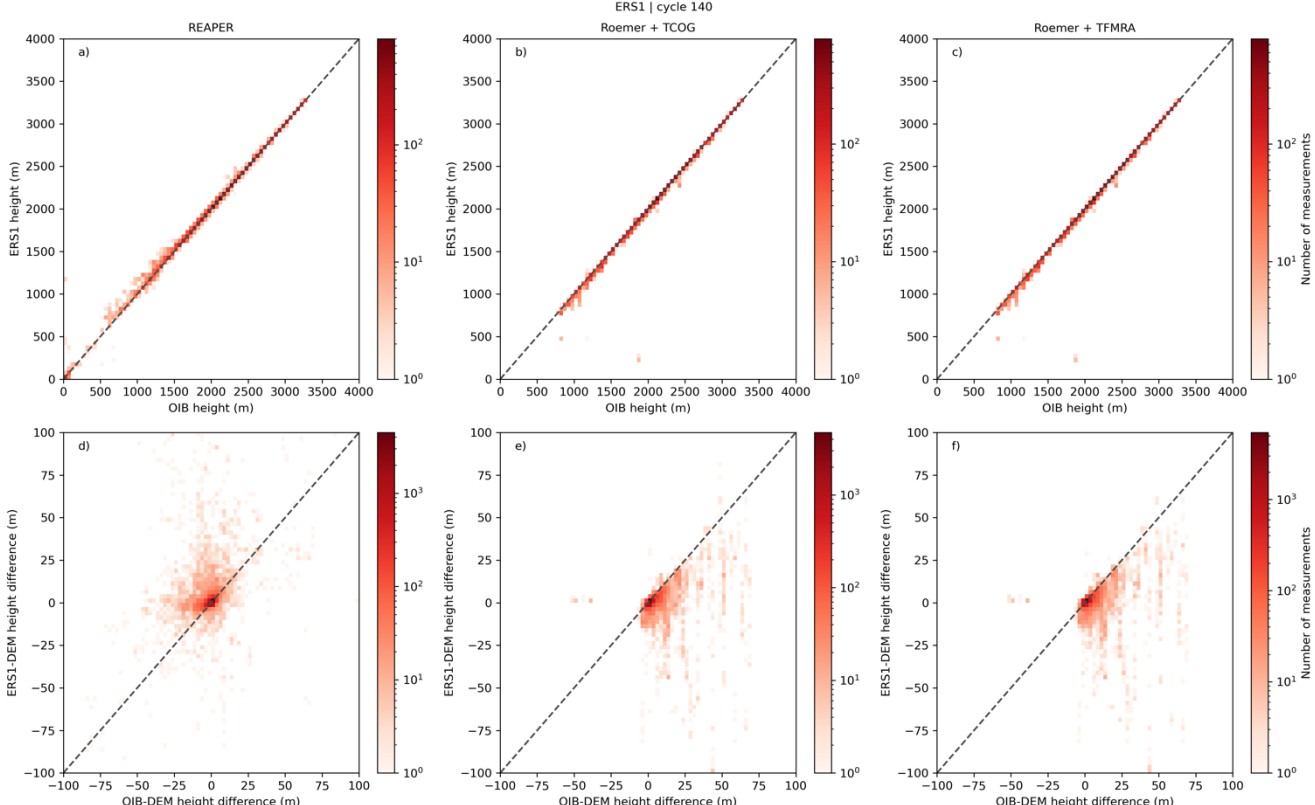

**Figure 11**. Density scatter plots showing the distributions of elevation measurements recorded by the ERS-1 and Operation IceBridge (OIB) airborne platforms. The top row (panels a-c) shows the original elevations; the bottom row (panels d-f) shows elevation residuals relative to an auxiliary DEM, which is used to remove the largescale topographic variance.

**Table 7.** Performance metrics summarizing the differences in elevation between ERS-1, ERS-2 and Envisat, and cotemporaneous airborne reference data. For each mission, statistics are provided for the baseline REAPER and Envisat version 3 products, together with the FDR4ALT TCOG and TFMRA retracking configurations. *Percentage of outliers* is defined as the percentage of measurements that deviate by more than 10 metres from the corresponding airborne measurement of elevation.

| Envisat | Version 3 | FDR4ALT TCOG | FDR4ALT TFMRA |
|---|---|---|---|
| Percentage of valid elevation measurements | 96.1 % | 93.6 % | 93.7 % |
| Number of comparison points | 8560 | 7534 | 7548 |




| | | | |
|---|---|---|---|
| Median elevation difference (m) | 2.25 | -0.51 | -0.49 |
| Median Absolute Deviation of the elevation differences (m) | 2.41 | 0.80 | 0.78 |
| Percentage of outliers | 34.5 % | 13.0 % | 14.3 % |

| ERS-2 | REAPER | FDR4ALT TCOG | FDR4ALT TFMRA |
|---|---|---|---|
| Percentage of valid elevation measurements | 91.6 % | 90.0 % | 92.0 % |
| Number of comparison points | 4778 | 4503 | 4504 |
| Median elevation difference (m) | 0.93 | -0.25 | -0.35 |
| Median Absolute Deviation of the elevation differences (m) | 0.87 | 0.84 | 0.89 |
| Percentage of outliers | 16.8 % | 9.3 % | 9.7 % |

| ERS-1 | REAPER | FDR4ALT TCOG | FDR4ALT TFMRA |
|---|---|---|---|
| Percentage of valid elevation measurements | 92.7 % | 91.1 % | 92.9 % |
| Number of comparison points | 12754 | 12255 | 12308 |
| Median elevation difference (m) | 0.17 | -0.81 | -0.36 |
| Median Absolute Deviation of the elevation differences (m) | 0.82 | 0.76 | 0.81 |
| Percentage of outliers | 13.8 % | 10.7 % | 10.7 % |

### 3.2.4 Influence of surface slope

In the previous analysis, we determined performance metrics that were aggregated at the ice sheet scale. Next, in order to
460    better understand the impact of surface topography, we assessed measurement accuracy as a function of ice sheet surface
slope. We performed the assessment for each satellite mission, in each case comparing the performance of the new
FDR4ALT datasets (TCOG solution) with the preceding baseline products (Figure 12). Whilst the performance is broadly
similar across the very lowest slopes of the interior of the ice sheet, it can be seen that the new FDR4ALT solution is much



more robust in the increasingly steeper sloped regions of the ice margin. For slopes greater than ~0.25°, the FDR4ALT
solution delivers increasingly significant reductions in both the median absolute bias and the dispersion of the elevation
differences relative to the airborne datasets. This is likely due to the implementation of a non-linear echo relocation
methodology (Roemer et al., 2007) within the FDR4ALT processor, which provides a more realistic estimate of the true
echoing point in areas of rugged topography, in combination with the FDR4ALT filtering and quality control. Comparing
performance across the three missions shows an apparent lower accuracy of Envisat at higher slopes, which is likely to
reflect the increased coverage that Envisat achieves in these regions (Figure 5).

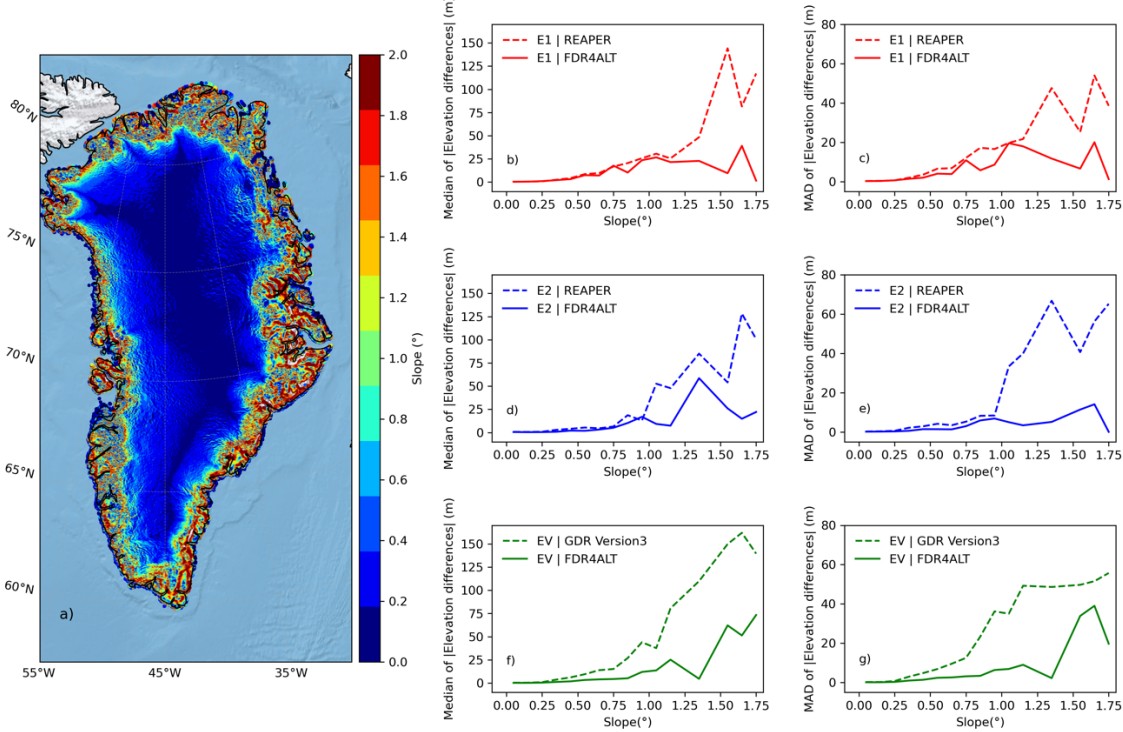

**Figure 12.** Comparison of baseline (REAPER and Envisat version 3) and FDR4ALT measurement accuracy as a function of
Greenland Ice Sheet surface slope, through comparison to airborne reference datasets. A retracking quality flag was applied
to all three products and, in the case of FDR4ALT, a relocation flag was also applied. a. Surface slope of the Greenland Ice
Sheet; b and c. the median absolute elevation difference (b) and the median absolute deviation (MAD) from the median of
the elevation differences (c) as a function of surface slope for ERS-1 REAPER (dashed line) and FDR4ALT TCOG (solid
line) datasets; d and e. the median absolute elevation difference (d) and the median absolute deviation from the median of the
elevation differences (e) as a function of surface slope for ERS-2 REAPER (dashed line) and FDR4ALT TCOG (solid line)
datasets; f and g. the median absolute elevation difference (f) and the median absolute deviation from the median of the
elevation differences (g) as a function of surface slope for Envisat version 3 (dashed line) and FDR4ALT TCOG (solid line).



## 3.3 Assessment of Waveform Morphology and Impact upon Measurement Accuracy

It is well established that variable surface topography and backscattering characteristics within the altimeter beam footprint impact the shape of the returned waveform, and cause divergence away from the classical Brownian-shaped echo. We

therefore used our neural network classification of waveform type, to investigate (1) how ERS-1, ERS-2 and Envisat waveform morphology varied over the Greenland and Antarctic ice sheets, and (2) the impact of waveform shape upon measurement accuracy.

### 3.3.1 Audit of Waveform Class

As described previously, our neural network classifier was used to distinguish a number of common classes of waveform

type. The proportion of waveforms within each class is summarized in Tables 8 and 9 and the spatial distributions of the five most common waveform types are shown in Figures 13 and 14, for Greenland and Antarctica, respectively. Across Greenland and Antarctica, ERS-1 and ERS-2 exhibit similar spatial distributions, which is unsurprising given the similarity of their instruments and modes of acquisition. Across the interiors of both ice sheets, ERS waveforms largely fall within classes 1, 6 and 13, corresponding to a Brownian-type echo, with or without an additional peak close to the leading edge.

The most notable divergence from this broad characterization occurs within the interior of East Antarctica, where ERS-2 shows a greater proportion of Brownian echoes without an additional peak, in comparison to ERS-1. Closer to the ice margin, the most dominant class for both ERS-1 and ERS-2 is class 13, indicating a Brownian-type shape with either a noisy leading edge or an indistinct trailing edge. In contrast, Envisat exhibits greater variance in the observed waveform shapes, potentially due to the prevalence of higher bandwidth acquisitions compared to ERS-1 and ERS-2 (Figure 1), which allows

more variability in the backscattered power to be resolved. Specifically, classes 1 and 7 are most common within the interior of both ice sheets, corresponding to classical Brownian-type echoes, and echoes with a less rapidly attenuating trailing edge. Moving towards the ice margins, Envisat waveform classes 9 and 11 become most dominant across both Greenland and Antarctica, corresponding to waveforms that have a very complex structure or a double or stepped leading edge.



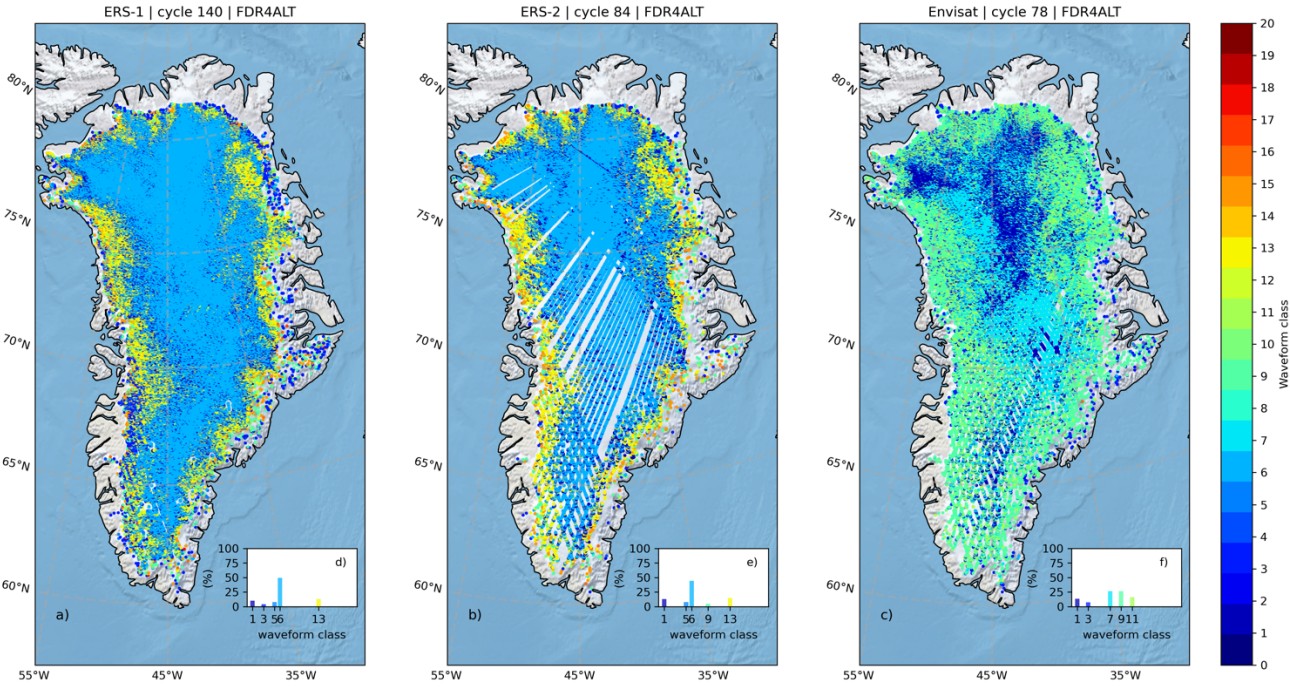


**Figure 13.** The spatial distribution of the five most common waveform classes over the Greenland Ice Sheet for one orbital cycle of ERS-1 (a), ERS-2 (b), and Envisat (c). In each of the main panels, data have been aggregated onto a 2 x 2 km grid, with the modal value of all data within a given cell shown. The inset figures show the percentage of grid cells covered by each of the top five waveform classes for each mission. The different waveform classes are defined in Tables 8 and 9.


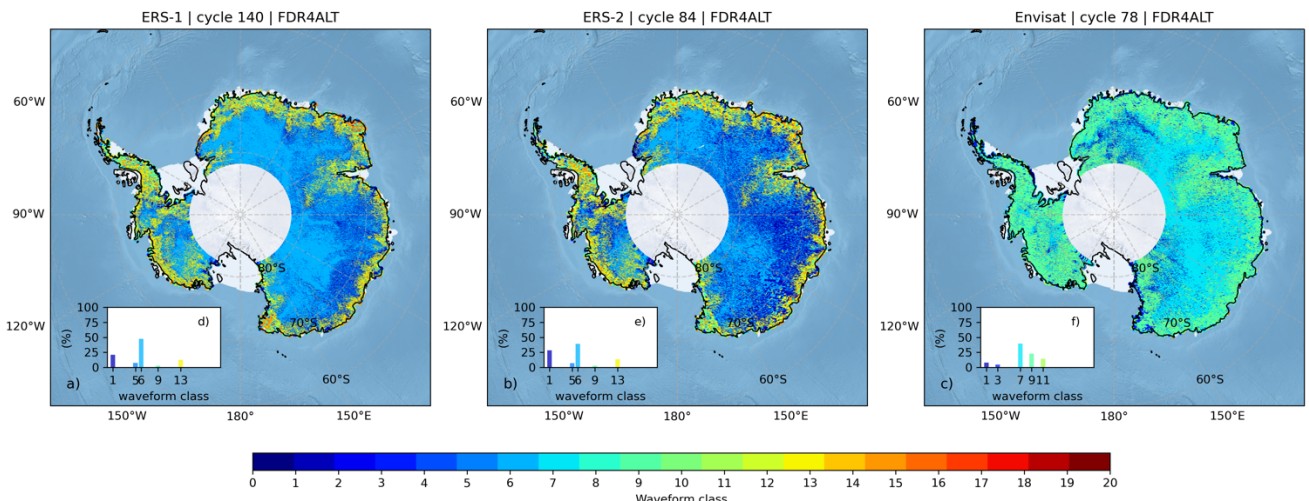



**Figure 14.** The spatial distribution of the five most common waveform classes over the Antarctic Ice Sheet for one orbital cycle of ERS-1 (a), ERS-2 (b), Envisat (c). In each of the main panels, data have been aggregated onto a 2 x 2 km grid, with the mode of all data within a given cell shown. The inset figures show the percentage of grid cells covered by each of the top five waveform classes for each mission. The different waveform classes are defined in Tables 8 and 9.

**Table 8.** The percentage of waveforms within each class across the Greenland Ice Sheet, for ERS-1, ERS-2 and Envisat data. The number in brackets indicates the 5 most common waveform classes, with 1 indicating the most prevalent class.

| Waveform class number | Description | ERS-1 (%) | ERS-2 (%) | Envisat (%) |
|---|---|---|---|---|
| 1 | Brownian | 10.29 (3) | 13.28 (3) | 13.61 (4) |
| 2 | Highly specular | 0.46 | 0.29 | 0.26 |
| 3 | Multiple peaks | 4.22 (5) | 2.78 | 7.52 (5) |
| 4 | Moderately specular | 0.63 | 0.34 | 0.06 |
| 5 | Brownian with a peak on the trailing edge | 8.02(4) | 7.95 (4) | 1.2 |
| 6 | Brownian with a peak on the leading edge or a steep trailing edge | 49.40 (1) | 44.41 (1) | 2.68 |
| 7 | Brownian with a flat trailing edge | 0.55 | 0.87 | 26.44 (2) |
| 8 | Strong peak at the end of the analysis window | - | - | 0.23 |
| 9 | Very complex echo | 4.09 | 5.13 (5) | 26.55 (1) |
| 10 | Brownian with high thermal noise | 1.07 | 1.26 | - |
| 11 | Double leading edge | 3.37 | 3.08 | 16.44 (3) |
| 12 | Shifted Brownian | - | - | 0.07 |
| 13 | Brownian with a disturbed leading edge | 13.26 (2) | 15.03 (2) | - |
| 14 | Volume-Brownian | - | - | - |
| 15 | Linear rise | 2.5 | 3.83 | - |
| 16 | Right-shifted Brownian waveform | 0.44 | 0.36 | 0.14 |
| 17 | Breakage on the leading edge of a Brownian waveform | 0.46 | 0.38 | - |
| 18 | Linear decrease | 1.16 | 0.93 | - |
| 19 | Small step before leading edge | - | - | - |





| 20 | Peaky echo before a Brownian waveform | - | - | - |


**Table 9.** The percentage of waveforms within each class across the Antarctic Ice Sheet, for ERS-1, ERS-2 and Envisat data. The number in brackets indicates the top 5 waveform classes by percentage, with 1 indicating the most prevalent class.

| Waveform class number | Description | ERS-1 (%) | ERS-2 (%) | Envisat (%) |
|---|---|---|---|---|
| 1 | Brownian | 20.73 (2) | 28.26 (2) | 7.69 (4) |
| 2 | Highly specular | 0.25 | 0.17 | 0.09 |
| 3 | Multiple peaks | 1.37 | 1.14 | 4.42 (5) |
| 4 | Moderately specular | 0.09 | 0.05 | 0.01 |
| 5 | Brownian with a peak on the trailing edge | 7.58 (4) | 6.98 (4) | 0.86 |
| 6 | Brownian with a peak on the leading edge or with a steep trailing edge | 47.26 (1) | 39.1 (1) | 2.03 |
| 7 | Brownian with a flat trailing edge | 0.72 | 1.14 | 39.56 (1) |
| 8 | Strong peak at the end of the analysis window | - | - | 0.13 |
| 9 | Very complex echo | 2.59 (5) | 2.85 (5) | 22.80 (2) |
| 10 | Brownian with high thermal noise | 1.15 | 1.24 | - |
| 11 | Stepped leading edge | 2.19 | 1.92 | 14.55 (3) |
| 12 | Shifted Brownian | - | - | 0.05 |
| 13 | Brownian with a disturbed leading edge | 12.64 (3) | 13.41 (3) | - |
| 14 | Volume-Brownian | - | - | - |
| 15 | Linear rise | 2.05 | 2.49 | - |
| 16 | Right-shifted Brownian waveform | 0.27 | 0.26 | 0.08 |
| 17 | Breakage on the leading edge of a Brownian waveform | 0.39 | 0.36 | - |
| 18 | Linear decrease | 0.64 | 0.35 | - |
| 19 | Small step before leading edge | - | - | - |



| 20 | Peaky echo before a Brownian waveform | - | - | - |
|---|---|---|---|---|

### 3.3.2 Impact of Waveform Class on Measurement Accuracy

Next, for each mission we evaluated the impact of waveform morphology upon the derived elevation accuracy, by partitioning the statistics from our accuracy assessment (Section 3.2) according to waveform class. This analysis allowed us to identify classes of waveforms that typically offer degraded performance using current processing approaches, and therefore where there may be scope to make algorithmic improvements in the future. Across all missions, waveform classes 1, 5, 6 and 7 show relatively high levels of accuracy (Figure 15), with median values close to zero and relatively low levels

of dispersion. These waveform classes correspond to broadly Brownian-type echoes, with some distortion to the classical shape due to a more prominent peak or more slowly decaying trailing edge. These phenomena arise due to the more complex nature of ice surfaces relative to ocean surfaces, with more variable footprint scale topography and greater penetration of the radar wave into the near surface snowpack. Nonetheless, the relatively good performance demonstrates the robustness of the empirical retracking and relocation approaches that have been implemented here. In contrast, relatively specular (classes 2

and 4) and multipeaked (class 3) waveforms generally produce a negative elevation bias, although the degradation in performance is less severe for Envisat than the preceding missions. For ERS-1 and ERS-2, the most common remaining classes are 9 (very complex) and 13 (disturbed leading edge). Whilst the accuracy of elevation measurements derived from class 13 waveforms is reasonable, class 9 waveforms exhibit a significant negative bias and large spread relative to coincident airborne data. For Envisat, beyond classes 1, 3 and 7, the most common remaining classes are 9 (very complex)

and 11 (stepped leading edge). In both cases, the accuracy of the associated elevation measurements remains relatively good, albeit with a small number of large outliers where the waveform complexity has not been adequately handled by the Level 2 processing. Overall, this analysis suggests that there may be scope to develop more sophisticated Level 2 processing approaches – namely retracking and relocation – that are specifically designed to handle complex, multi-peaked waveforms; and that this could yield further improvements in measurement accuracy over complex topographic regions in the future.



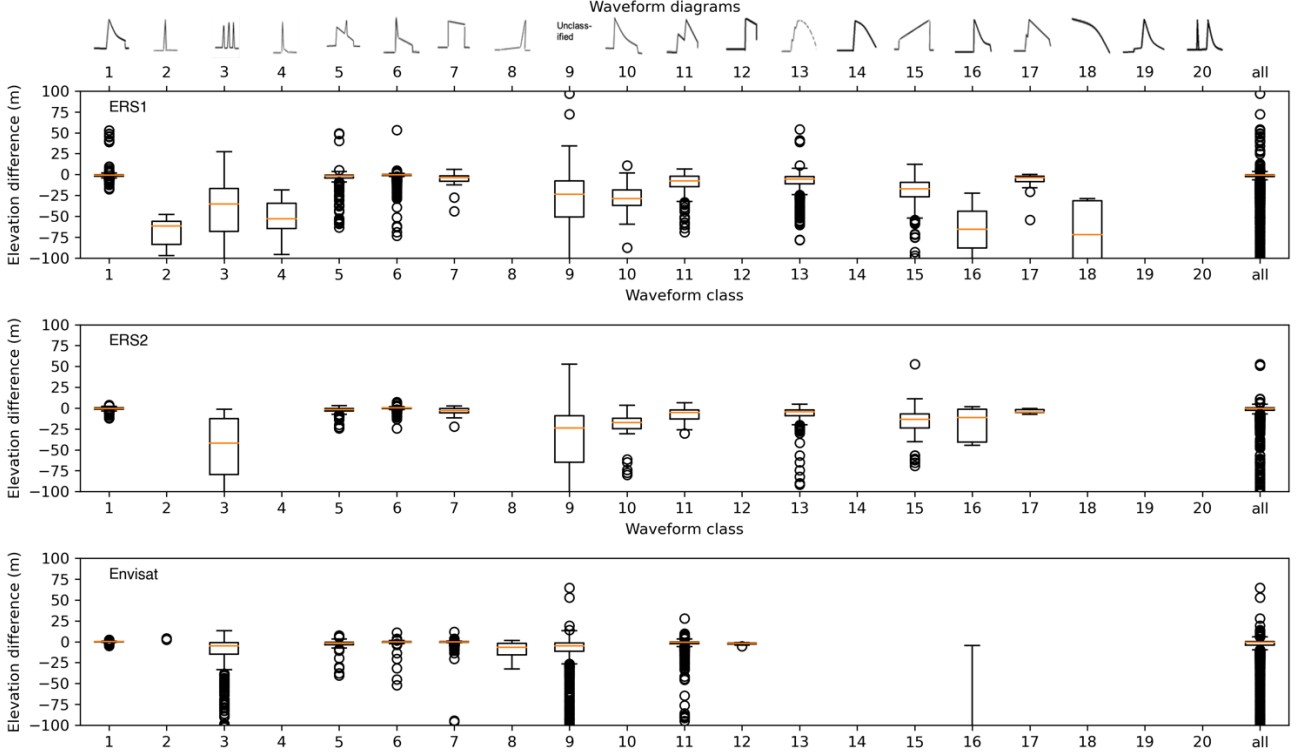


**Figure 15.** Box and whisker plot representing the distributions of elevation difference (mission-minus-airborne) for each of the waveform classes obtained from the neural network classification, for ERS-1 (upper panel), ERS-2 (middle panel) and Envisat (lower panel). On each plot, the orange line denotes the median elevation difference, the boxes represent the lower and upper quartiles, the whiskers indicate the range spanned by 99% of the data, and the circles locate the remaining outliers.

Classes 10, 13-15, 17-20 and 8, 12, 14, 19-20 are excluded respectively for ERS and ENVISAT mission as they are not defined.

### 3.4 Thematic Data Product Assessment

As described previously, one of the principal objectives of the TDP is to produce a more consistent product through time, by correcting for topographically-induced elevation differences resulting from the orbital drift of the satellite. As such, we

evaluate the TDP elevation measurements by assessing their stability through time. Ultimately, this is beneficial for the reliable determination of ice sheet evolution, and therefore this assessment allows us to determine the extent to which the TDP processing chain has improved upon the existing Level-2 product in terms of delivering a more consistent dataset, particularly for the non-altimetry expert user. More specifically, our assessment of the TDP was performed by computing the standard deviation of elevations across orbit cycles, at defined intervals along satellite tracks. The assessment was performed

for all tracks crossing Greenland and Antarctica, and for each of the three missions. In the case of ERS-1, we computed the



standard deviation for only data acquired whilst in its 35-day orbit, to ensure that it most closely matched the orbital configuration of ERS-2 and Envisat, and thus provided consistency across the three satellite missions. In all cases the metrics were computed for both the FDR4ALT Level 2 and TDP products, to assess the impact of the additional TDP processing. This analysis shows that the TDP achieves a much lower standard deviation in elevation, in comparison to the Level 2 product, for both Greenland and Antarctica and for all missions (Figure 16, Figure 17 and Figure 18 for ERS-1, ERS-2 and Envisat, respectively).

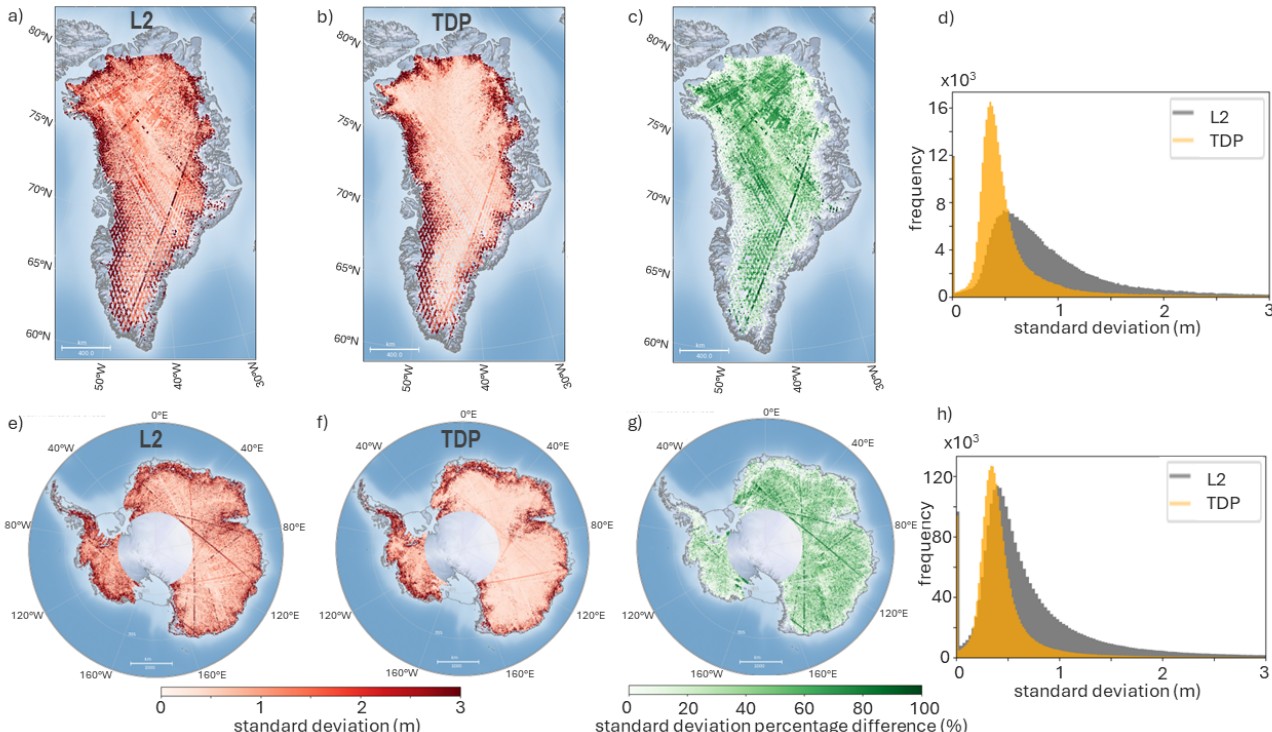

**Figure 16.** Assessment of the internal stability of the ERS-1 FDR4ALT Land Ice Level 2 and TDP products for tracks over the Greenland (a-d) and Antarctic (e-h) (row 1) ice sheets showing the temporal standard deviation for the Level-2 product (a and e), the temporal standard deviation for the TDP product (b and f); the percentage improvement in standard deviation between the Level-2 and TDP products (c and g); and the distribution of standard deviation for all tracks, for Greenland (d) and for Antarctica (h), for the Level-2 product (grey) and the TDP product (orange).



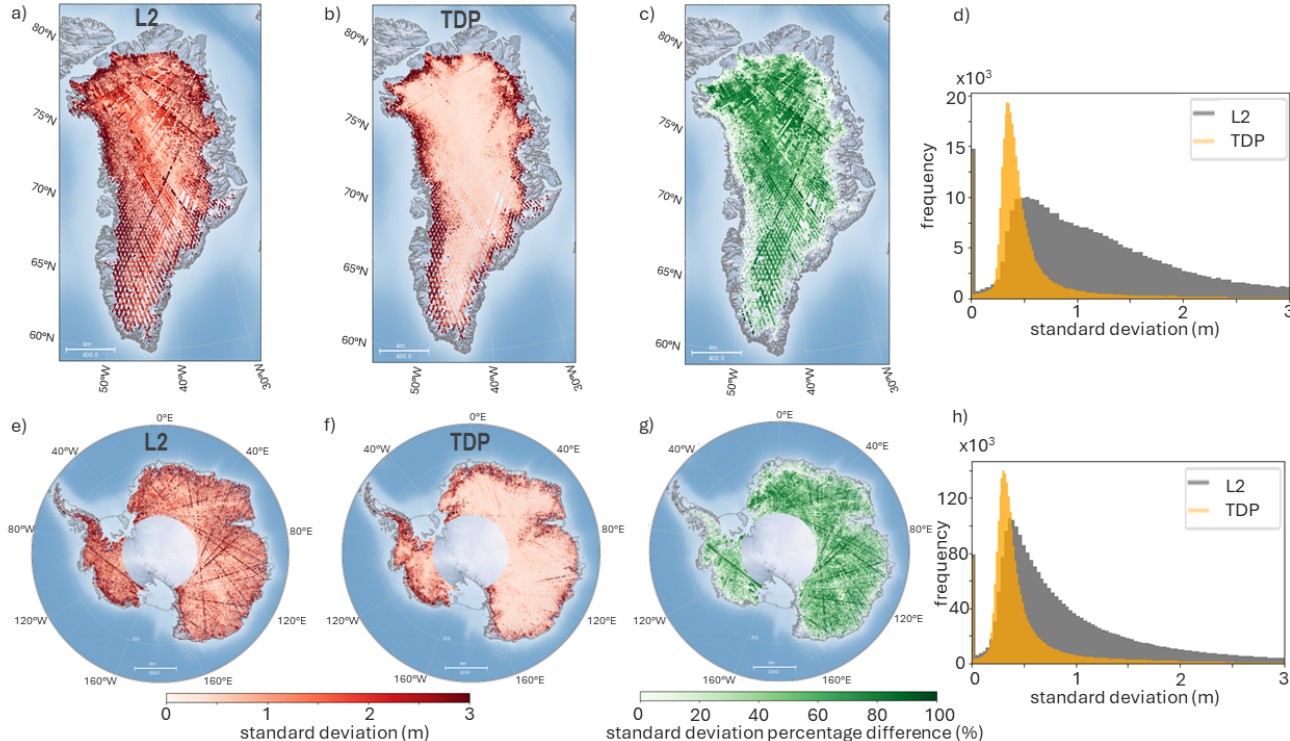

**Figure 17.** Assessment of the internal stability of the ERS-2 FDR4ALT Land Ice Level 2 and TDP products for tracks over the Greenland (a-d) and Antarctic (e-h) ice sheets showing the temporal standard deviation for the Level-2 product (a and e), the temporal standard deviation for the TDP product (b and f); the percentage improvement in standard deviation between the Level-2 and TDP products (c and g); and the distribution of standard deviation for all tracks, for Greenland (d) and for Antarctica (h), for the Level-2 product (grey) and the TDP product (orange).





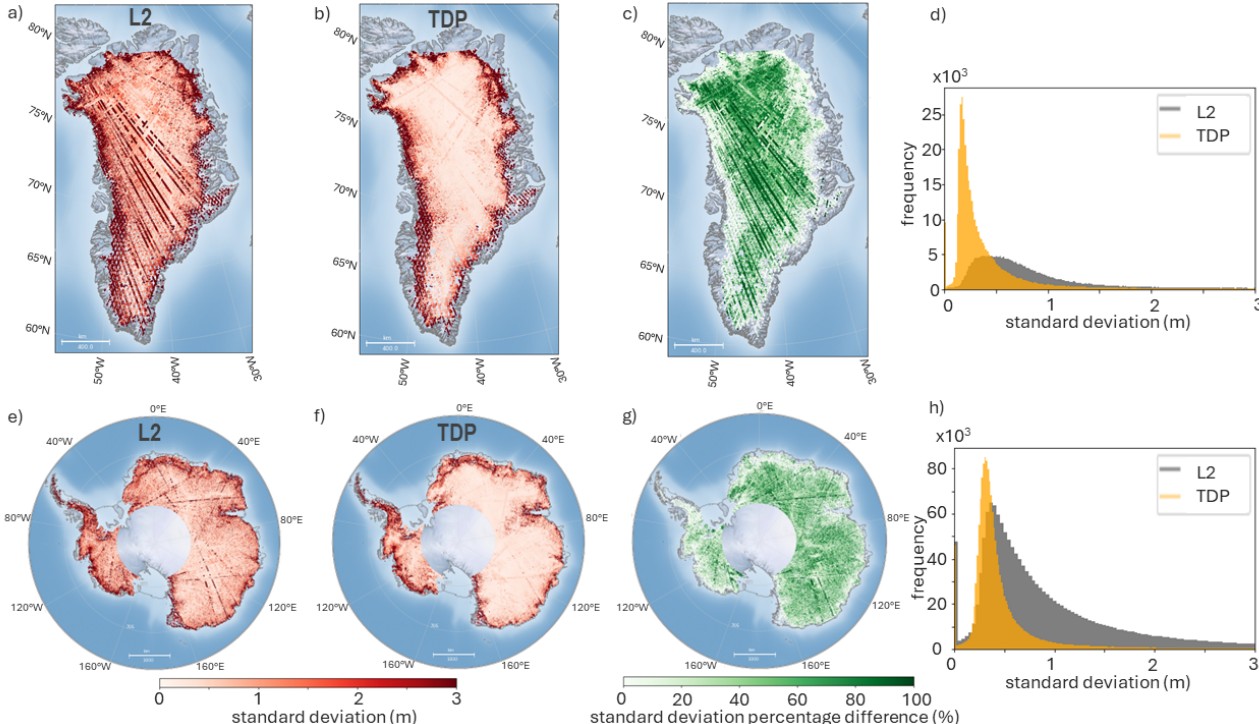

**Figure 18.** Assessment of the internal stability of the Envisat FDR4ALT Land Ice Level 2 and TDP products for tracks over the Greenland (a-d) and Antarctic (e-h) ice sheets showing the temporal standard deviation for the Level-2 product (a and e), the temporal standard deviation for the TDP product (b and f); the percentage improvement in standard deviation between the Level-2 and TDP products (c and g); and the distribution of standard deviation for all tracks, for Greenland (d) and for Antarctica (h), for the Level-2 product (grey) and the TDP product (orange).

## 4 Conclusions

In this study, we have presented a new reprocessing of the ERS-1, ERS-2 and Envisat radar altimetry datasets over the ice sheets of Greenland and Antarctica. The reprocessing ingests Level-1b data, applies updated Level-2 processing that is tailored to ice sheets, and adds additional Level-2+ algorithms that generate for the first time a Thematic Data Product, which is designed to be more accessible for the non-expert user. We perform a comprehensive assessment of the accuracy of these new datasets by comparing them to contemporaneous airborne measurements, and evaluate changes in performance relative to the existing REAPER and Envisat version 3 baseline products. Overall, we find that the updated processing leads to a closer agreement with airborne data, both in terms of the median bias and the dispersion of the differences. As part of the analysis, we compare results from two empirical retrackers and find only small differences in performance between the two. As such, we conclude that updates in other Level-2 processing steps, such as the algorithms used for echo relocation and quality control exert a larger influence on overall measurement accuracy. We implement a neural network classifier to



explore the different classes of waveform shape present over the ice sheets, and the extent to which measurement accuracy varies as a function of waveform morphology. The datasets generated in this study will be made publicly available by the Europe Space Agency, and provide the opportunity for improved long-term constraint of ice sheet elevation change, mass imbalance and, ultimately, a better understanding of their contribution to sea level rise.

**Author contributions**

MM conceived and supervised the study. MS, JM, FP, JA, QH, JAD and CG processed the data and performed the experiments. JAD developed and validated the AI algorithm. MS, MM, JM, FP, JA and JAD drafted the manuscript. All authors contributed to the analysis of the results and commented on the manuscript.

**Competing interests**

The contact author has declared that none of the authors has any competing interests.

**Acknowledgments**

This study was funded by the European Space Agency Fundamental Data Records for Altimetry project, under contract N°4000128220/19/I-BG. MM was supported by the UK Centre for Polar Observation and Modelling (grant no. cpom300001) and the Lancaster University-UKCEH Centre of Excellence in Environmental Data Science. The Level-2 processing was performed using resources from the CNES High Performance Computing Center.

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
