# Peer review of "New Radar Altimetry Datasets of Greenland and Antarctic Surface Elevation, 1991-2012"

_EGUsphere, 2024_

## Author Response (AR1)

**Response to Reviewers**

We thank the reviewers and the editors for their comments. We address each of the points raised below, with our response provided after each comment.

Referee #1
The manuscript reports the generation and assessment of new data products of Greenland Ice Sheet and Antarctic Ice Sheet surface elevation from the ERS-1, ERS-2 and Envisat radar altimetry missions. The underlying reprocessing has been performed within ESA's Fundamental Data Records for Altimetry (FDR4ALT) project. It is to supersede the previous REprocessing Altimeter Products for ERS-2 and ERS-2 (REAPER) and the version 3 products for Envisat.

The data products are on the Level-2 level and contain position and surface elevation along track of satellite measurements. In addition a Thematic Data Product (TDP) is provided (which could be classified as Level-2b). It refers all measurements to reference ground tracks and thereby corrects for the non-exact-repeat nature of the orbits and measurements and for the associated topography-induced variations. I expect that this will largely facilitate higher-level analyses for temporal surface elevation change.

The improvements in the processing methodology include up-to-data retracking methods and the Roemer et al. 2007 approach of accounting for local topography by relocation.

The new data products, as well as the previous REAPER and Envisat version 3 products are extensively assessed and validated. Comparison to independent surface elevation data from Operatoin ICEBridge is done for Greenland. For both ice sheets, and for the three missions, waveforms are classified (using 21 different classes) through a Neural Network classification. The geographic distribution and histograms of the waveform classes are shown. For Greenland, this allows to assess surface elevation accuracy per waveform class, with very interesting insights and hints to future work on the waveform classes associated with low accuracy. It is shown convincingly that the new products outperform the previous products in virtually all aspects considered.

I enjoyed reading this manuscript. It is extremely clear and informative, starting with the helpful and succinct summary of the three missions in Section 2.1.-2.3 through the illustration on how the TMP simplifies variability in Section 3.4. Thank you for making the review work so easy.

Response: Thank you for these comments, we appreciate your recognition of the care we put into preparing the manuscript.

I only have a few minor-to-moderate comments on how the presentation could be further improved.

Comment 1: From the abstract and even from the introduction it is not clear what kind of dataset is presented. For example, readers of a certain background might expect a gridded data product. So you might specify that it is a Level-2 dataset comprising horizontal coordinates (lat lon, or some projection?) and surface elevation (ellipsoidal heights?) sampled along the satellite track.

Response 1: We acknowledge that this was not sufficiently clear, and have added the requested details. Specifically, we have mentioned in the abstract that these are along-track datasets, and also added text to the introduction to specify that these are L2 and L2+ products comprising elevation measurements sampled along the satellite track.

Comment 2: The manuscript presents datasets (rather than cryospheric science results). Therefore, it would be really appropriate to provide public access to the datasets at the time of publishing the paper, at latest. Currently, the manuscript does not comply with TC data policy (https://www.the-cryosphere.net/policies/data_policy.html). One might even argue that a data journal like ESSD might be more appropriate for this work. Anyway, I would not mind to see this work published in "The Cryosphere" because the excellent analysis of the data (such as the dependence of accuracy on slope and waveform classes) really facilitates understanding the nature of altimetric surface elevation measurements. And this is the basis for a lot of science.

Response 2: We agree with these points, and confirm that the data are now publicly available and distributed by ESA. Further details on how to access the datasets can be found at https://earth.esa.int/eogateway/catalog/tdp-for-land-ice. We have updated the manuscript to make this clear. Regarding our wish to publish this work in The Cryosphere rather than ESSD, we agree that in principle it would be in scope for both journals. At the stage of initial submission, we discussed this amongst all coauthors, and decided that we preferred to submit to the former for the reasons alluded to by the reviewer. We are therefore happy to see that the reviewer appreciates the appropriateness of this approach.

Comment 3: On a similar line, the reader is referred to technical documents for details of the processing (The FDR4ALT Detailed Processing Model Document; the FDR4ALT Product User Guide), which are currently available at websites. It would be desirable to see some commitment for their persistent availability. (It's also curious that these documents have "confidential" at the bottom of every page.). I'm not going to insist on this comment, though, as I understand it may not be trivial.

Response 3: We can confirm that these documents are now stored on the ESA website, as is the norm for technical documentation provided by space agencies. In view of this comment, we have additionally confirmed with ESA that they do indeed have policies and processes in place, such that the long term preservation and availability of this information and documentation is guaranteed. We have updated the links in the references section of the revised manuscript, to reflect the locations of these documents.

**Comment 4:** The abstract could be a bit more specific as to what kind of improvements have been implemented in the processing and what improvements could be achieved w.r.t. previous datasets.

**Response 4:** As requested, we have revised the abstract to include more specifics.

**Comment 5:** Concerning the validation by ICEBridge data, the reader might wonder why this exercise was restricted to Greenland. Maybe you can explain or justify this limitation in a sentence or two.

**Response 5:** As requested, we have added additional details to explain the justification for this approach. We selected Greenland specifically for validation purposes because (1) it provided the most extensive coverage of validation data for all missions, most notably for pre-IceBridge campaigns, where coverage across Antarctica is extremely limited, and (2) it encompassed a broad range of topographic complexity and surface backscattering characteristics. We believe that in doing so, our analysis provides a relatively comprehensive assessment of data accuracy, and indeed goes far beyond the level of validation undertaken for previous products (e.g. Brockley et al., 2017). We have updated the manuscript to make these points clear.

Brockley, D. J., Baker, S., Femenias, P., Martinez, B., Massmann, F. H., Otten, M., Paul, F., Picard, B., Prandi, P., Roca, M., Rudenko, S., Scharroo, R., & Visser, P. (2017). REAPER: Reprocessing 12 Years of ERS-1 and ERS-2 Altimeters and Microwave Radiometer Data. *IEEE Transactions on Geoscience and Remote Sensing*, *55*(10), 5506 - 5514. https://doi.org/10.1109/TGRS.2017.2709343

**Comment 6:** line 84 "had the impact of" --> "had the effect of"

**Response 6:** Agreed; we have updated the manuscript as suggested.

**Comment 7:** line 704 something wrong about this reference "Team, I"

**Response 7:** Agreed; this has been corrected.

**Comment 8:** line 702 same problem

**Response 8:** Agreed; this has been corrected.

**Comment 9:** In the discussion of the relocation you might mention that the geographic position assigned to the measurement is corrected together with the height. This is (of course) not reflected in Eq. 1.

**Response 9:** We have added this point, as requested.

Comment 10: line 187: As fas as I understand, the ocean tide and inverse barometer effect are zero over ice sheets. This could be mentioned for completeness.

Response 10: Yes, this is correct; we have added text, as requested, to clarify this point.

Comment 11: 289 "mean square error". Maybe "mean square difference" might be a more appropriate term.

Response 11: Agreed; we have changed this as suggested.

Comment 12: line 292ff: The reader learns that this classification was performed for waveforms from not only over the ice sheet.

Response 12: Yes this is correct; we have added text to clarify this point and the motivation for adopting this approach.

Comment 13: Make sure in the text that the reader understands which steps are done for global altimetry data and which steps are done specifically for ice sheets.

Response 13: As requested, we have clarified where any steps have been performed on the global altimetry dataset, as opposed to specifically for ice sheets.

Comment 14: line 345 "less continuous along-track sampling". I think you could sell this point more positively and avoid the impression (at first reading of this sentence) that this is some disadvantage.

Response 14: As suggested, we have reworded this more positively.

Comment 15: line 374 replace "Threshold Centre of Gravity" by TCOG for consistency and brevity.

Response 15: Agreed; we have changed this as requested.

Comment 16: Fig. 6: This is a great visualisation. Just the histogram insets are too small to be well readable, at least in a printed version. You may try to make them a bit bigger. And convince the journal in the layout process to use the full paper width for the figure. In turn you might cut the headline "Envisat | cycle 78" and put this information into the figure description.

Response 16: Thanks for your positive comments about this figure. As requested, we have tried to increase the histograms' size as much as possible, whilst not detracting from the overall figure by obscuring the main panels. We have also removed the headline "Envisat | cycle 78" and put this information in the caption, as suggested.

Comment 17: Somehow, Figure 10 is missing completely.

Response 17: We apologise for this omission and we have added the figure in the revised version.

Comment 18: Fig. 12a: It appears that slopes are shown not only for the ice sheet area, which is inconsistent with what is sayed in the figure caption and with the other figures in the manuscript.

Response 18: Thanks for spotting this; as requested, we have updated the figure so that the slopes are now masked to the ice sheet area only.

Comment 19: line 490f "Comparing performance across the three missions shows an apparent lower accuracy of Envisat at higher slopes, which is likely to reflect the increased coverage that Envisat achieves in these regions (Figure 5)." I suggest to replace "Comparing performance" by "Comparing the performance of FDR4ALT" in order to be cristal-clear.

Response 19: Agreed; we have updated the manuscript according to this suggestion.

Comment 20: I suggest the reference to Fig. 5 should be replaced by a reference to Table 7, because the increased coverage cannot easily seen from Fig. 5, while it is clearly quantified in Table 7

Response 20: Agreed; we now refer to Table 7 instead of Figure 5.

Comment 21: Fig. 12 caption: I suggest to shorten the description by writing something like:

"d and e: same as b and c, but for ERS-2. f and g: same as b and c, but for Envisat"

Response 21: We have shortened the caption, as suggested.

Comment 22: Fig 13 and 14. The caption says "The spatial distribution of the five most common waveform classes". However it seems that not only the five most common classes are depicted. For example, panel a and b show a lot of orange dots, too.

Also, the the color bar is somewhat ambiguous in assigning colors to waveform classes. (Maybe stretch it from -0.5 to 20.5, then the ticks at 0, 1, ..., 20 are in the centre of their color block)

Response 22: Thanks for spotting this; as requested we have corrected the captions of these figures, and also modified the colour bars so that the ticks and labels are located at the centre of each colour block.

Comment 23: Congratulations for Figure 15. I like your graphical indication of waveform classes at the top of the figure!

Yet I have some minor comments on this figure.

- Choose one name for class 9: either "very complex echo" (as in Tab. 2, 8, 9) or "unclassified" as in the top of Fig. 15.

- There is maybe too much graphical emphasis on the outliers. - certainly a matter of taste.

- Figure 15 caption, last sentence: I guess the order must be changed into "Classes 8, 12, 14, 19-20 and 10, 13-15, 17-20" to match the order "ERS and Envisat"
Throughout the paper, be consistent in writing Envisat (rather than ENVISAT)

Response 23: Thanks, we are glad that you like this figure. As requested, we have (1) replaced "unclassified" with "Very complex echo" in the top label of the figure; (2) revised the figure to place less graphical emphasis on the outliers, by marking them as small dots rather than open circles; and (3) corrected the ordering of the classes in the caption to match the order of 'ERS and Enivsat'. We have also checked the manuscript to ensure that we consistently use 'Envisat' throughout.

Comment 24: line 534ff "In contrast, relatively specular (classes 2 and 4) and multipeaked (class 3) waveforms generally produce a negative elevation bias, although the degradation in performance is less severe for Envisat than the preceding missions."

The second half of this sentence is substantiated for class 3 but not for classes 2 and 4. Maybe limit this consideration to class 3, as occurance of class 2 and 4 is rare.

Response 24: Agreed, thanks for pointing this out; we have limited this statement to class 3 only, as suggested.

Comment 25: line 539 "For Envisat, beyond classes 1, 3 and 7, the most common remaining classes are 9 (very complex) and 11 (stepped leading edge)."

This sentence seems to suggest that classes 9 and 11 are less common than classes 1, 3, 7, but this is not the case.

Response 25: We agree that our previous wording was ambiguous here, and so we have modified the text to make this statement clearer.

Comment 26: Fig. 16 caption: in line 571, "(row 1)" does not appear to make sense to me.

Response 26: Agreed, we have removed '(row 1)' as it is redundant given that the panels are labelled.

Referee #2

This paper presents an important contribution to the field of ice sheet monitoring by improving the accuracy and consistency of satellite radar altimetry data from ERS-1, ERS-2, and Envisat. The research is highly relevant given the significant acceleration in ice loss from the Greenland and Antarctic Ice Sheets over the past three decades.

A key strength of the study is its focus on refining data from older historical satellite missions. By optimizing retrieval methods and evaluating measurement accuracy through comparisons with independent airborne datasets, the authors provide a more reliable multi-decadal record of ice sheet elevation changes. This effort enhances the ability to assess long-term trends in ice sheet imbalance, placing contemporary observations in a broader temporal context.

The study is well-motivated and addresses a critical gap in the accurate long-term monitoring of ice sheets. The findings have wide-ranging applications, including improving estimates of ice mass loss. The comprehensive validation of the new datasets further strengthens the credibility of the results.

Response: Thank you for these positive comments.

Comment 1: However, a more detailed discussion of the specific retrieval methods used and how they compare to prior approaches would enhance the clarity of the study's technical advancements.

Response 1: As requested, we have added further detail, where relevant, of the various retrieval methods, together with information relating to how the approaches implemented within this study differ from prior processing strategies. Additionally, we also provide references to technical project documentation now hosted by ESA, which provides more comprehensive algorithmic detail and validation, should it be of interest to the reader.

Comment 2: Additionally, insights into the potential limitations of the new datasets and their implications for future research would provide a more balanced perspective. Including a section on the limitations of the new data product would strengthen the manuscript.

Response 2: Thanks; we agree with this suggestion. As requested, we have therefore added a completely new section to address Limitations and Opportunities for Future Research, which we believe provides a useful addition to the manuscript.

Comment 3: It is important for reviewers to have access to the new dataset during the review process. I find the following statement confusing: "The datasets generated in this study will be made publicly available by the European Space Agency…" When will the data be available? next year? The authors could consider providing a private link for review purposes. At this moment, I am unable to provide feedback on the data (file format, structure, uncertainties, etc.). Since this paper is a data paper, I find it strange that the data is not already available.

Response 3: We apologise for this oversight, and can confirm that the data are now publicly available and distributed by ESA, as the entity who funded the project. Further details relating to data access can be found at https://earth.esa.int/eogateway/catalog/tdp-for-land-ice. We have updated the manuscript to make this clear.

Comment 4: EGUsphere is not a data journal (unless I am mistaken), unlike ESSD, for example. To justify publication in EGUsphere, the authors should present novel scientific insights. For instance, they could provide an ice volume estimate using the new and improved data product or compare it with other ice loss studies and highlight the differences. This will also emphasize the importance of the new data.

Response 4: Please also see the related comment by reviewer 1, who was happy that our analysis was well aligned with the scope of The Cryosphere. We also note your comment above, that the manuscript in its current form presents an important contribution to ice sheet monitoring, which we believe also supports its relevance to The Cryosphere. Additionally, we emphasise our view that deriving accurate measurements of ice sheet elevation in itself is important for a range of cryospheric applications, and that it is not just dh/dt and dv/dt that are relevant to the Cryosphere community. We realise that this last point was not particularly clear in the original manuscript, and so we have added additional text within the introduction to address this.

Beyond this, we have carefully considered this comment; however to provide the analysis suggested above would represent a major addition and refocus of the manuscript. Namely it would involve adding new sections on deriving dh/dt and dv/dt to the methods section of the manuscript; adding new sections detailing these additional dh/dt and dv/dt results; comparing these results to existing ice loss studies and undertaking detailed analysis to assess the impact of differences in the methodologies used to compute dh/dt and dv/dt etc, as well as the input data themselves; and developing historical dh/dt reference datasets for validation, in order to reach useful conclusions about any performance differences.

Given (1) the amount of work that this would involve, (2) the resulting shift in the existing focus of the manuscript from elevation to instead documenting and assessing estimates of volume change, and (3) the fact that the manuscript is already rather long and the analysis quite extensive, we believe that the above suggestion is beyond the scope of the current manuscript, and instead would be better suited to being addressed within the context of future work. Given the overall comments of the reviewers, we believe that the current scope and content of the manuscript is both relevant, and of interest, to The Cryosphere community as it stands.

Comment 5: Airborne campaign data is used to validate the new ERS-1, ERS-2, and Envisat products. Why not use airborne campaign data from 2002 to 2012? Figure 6 shows elevation differences (Envisat minus airborne) during March–May 2009, but does this change over time?

Response 5: As noted, we have used airborne campaign data to validate each of the new ERS-1, ERS-2 and Envisat products, and to perform comparison to previous products, namely REAPER and Envisat version 3. In choosing validation cycles, we selected those where there was the

greatest coverage of airborne data. Whilst we acknowledge that it would be possible to also use data from other cycles where the airborne coverage was lower, we note that our analysis already goes far beyond the level of validation previously performed (e.g. Brockley et al., 2017) and provides a relatively extensive validation test scenario in its current state, and thus a comprehensive assessment of the quality of the data. Hence we do not believe it would add significantly to the results to run the analysis with additional cycles with lower airborne coverage. Regarding the specific point about attempting to constrain changing elevation differences through time – due to the highly heterogenous sampling provided by airborne campaign data, resulting from the varying flight plans in different years, we do not believe it would be feasible to do this with confidence; i.e. to separate sampling effects from real change. We have added additional text to the manuscript to make our reasoning for these decisions clearer.

Brockley, D. J., Baker, S., Femenias, P., Martinez, B., Massmann, F. H., Otten, M., Paul, F., Picard, B., Prandi, P., Roca, M., Rudenko, S., Scharroo, R., & Visser, P. (2017). REAPER: Reprocessing 12 Years of ERS-1 and ERS-2 Altimeters and Microwave Radiometer Data. *IEEE Transactions on Geoscience and Remote Sensing*, *55*(10), 5506 - 5514. https://doi.org/10.1109/TGRS.2017.2709343

Comment 6: Figure 1: In panel 1b, the areas with no data are difficult to discern.

Response 6: As requested, we have updated the colour map so that it is easier to distinguish the areas with no data.

---

## Author Response (AR2)

**Comment 1:** The authors have done a great job responding to the referees, and the revised manuscript looks to be in very good shape.

**Response 1:** Thank you for these comments.

**Comment 2:** I am ticking the 'Publish subject to technical corrections' because I'd like to request two minor changes to the figures:

**Response 2:** Thank you for your approval. We have made the two minor figure changes as requested.

**Comment 3:** The image quality in the depiction of the waveform classes in figures 2 and 15 is not very good. This may be something that will be remedied during production of with the submission of full resolution figures, but I don't want it to be missed.

**Response 3:** We apologise for this. We have regenerated higher quality versions of the figures in Table 2 and Figure 15.

**Comment 4:** It is cumbersome for the reader to cross reference between table 9 and figure 15. Please include the waveform percentage for each class in figure 15. It would be acceptable to do this for only the four most prevalent classes for each altimeter if that simplifies the layout.

**Response 4:** Agreed. As requested, we have modified Figure 15 to include the five most prevalent classes for each altimeter, to match the number ranked in Table 8.

Thanks for your hard work on this!